# GWAS meta-analysis of psoriasis identifies new susceptibility alleles impacting disease mechanisms and therapeutic targets

Psoriasis is a common, debilitating immune-mediated skin disease. Genetic studies have identified biological mechanisms of psoriasis risk, including those targeted by effective therapies. However, the genetic liability to psoriasis is not fully explained by variation at robustly identified risk loci. To refine the genetic map of psoriasis susceptibility we meta-analysed 18 GWAS comprising 36,466 cases and 458,078 controls and identified 109 distinct psoriasis susceptibility loci, including 46 that have not been previously reported. These include susceptibility variants at loci in which the therapeutic targets IL17RA and AHR are encoded, and deleterious coding variants supporting potential new drug targets (including in *STAP2*, *CPVL* and *POU2F3*). We conducted a transcriptome-wide association study to identify regulatory effects of psoriasis susceptibility variants and cross-referenced these against single cell expression profiles in psoriasis-affected skin, highlighting roles for the transcriptional regulation of haematopoietic cell development and epigenetic modulation of interferon signalling in psoriasis pathobiology.

Psoriasis is a common immune-mediated skin disease with significant impact on psychosocial wellbeing, lifelong morbidity, and mortality[1,2]. With an estimated 60 million people affected worldwide[3], it represents a substantial economic burden[4,5].

Psoriasis has a strong genetic component, with heritability estimated at 66%[6]. Previous genome-wide association study (GWAS) meta-analyses have identified 65 genomic loci at which genetic variation is associated with psoriasis susceptibility in European ancestry populations[7–9], and 17 more reported in Asian ancestry populations[10]. A substantial fraction of genetic risk is attributed to the Major Histocompatibility Complex (MHC) class I allele HLA-C*06:02 and related antigen processing and presentation functions, while pathogenic roles for the IL-23/IL-17 immune axis, type I interferons and NF-κB have also been established[7,11–13]. Evidence that genetic variation influences psoriasis risk through these pathways underscores the remarkable consistency between genetic perturbations and the effectiveness of biologic therapies targeting IL-23 and IL-17[1].

While they have greatly influenced current models of psoriasis pathobiology, previous GWAS meta-analyses of psoriasis have been modest in size compared to those of other common diseases in the current era of population-based bioresources[14]. Larger studies offer enhanced ability to detect genetic associations with small effects and discriminate independent association signals within loci. There remain open questions around how, and in which cell types, the presence of risk-increasing alleles can lead to the dysregulated immune processes that characterise psoriasis. Unpacking these causal mechanisms and better understanding their heterogeneity across individuals should help inform how existing targeted therapies can be deployed to specifically disrupt the inflammatory loop underlying psoriasis while limiting unintended adverse consequences[15,16], and suggest new therapeutic targets for patients in whom current treatments are ineffective or response is transient[17]. It will also begin to explain the mechanistic basis for the high burden of co-morbidities suffered by individuals with psoriasis[18,19].

Here we report a meta-analysis of 18 case-control genome-wide association studies conducted by an international consortium to increase the statistical power for genetic discovery in psoriasis, and to characterise causal variants, genes, pathways, and cell types.

✉e-mail: michael.simpson@kcl.ac.uk; jelder@umich.edu

## Results

### Discovery of new psoriasis susceptibility regions

To identify genomic loci at which genetic variation is associated with psoriasis susceptibility, we performed fixed-effect standard-error-weighted GWAS meta-analysis for a total of 11,808,957 autosomal variants across 18 studies comprising a total of 494,544 unrelated European ancestry individuals. The genomic inflation factor[20] ($\lambda_{GC}$) of 1.14 and LD score regression intercept of 1.07 indicate modest inflation of the meta-analysis test statistics that is primarily driven by polygenicity (estimated proportion ascribed to other causes: 0.15), consistent with other complex diseases[21].

Consistent with previous GWAS of psoriasis, by far the strongest evidence of association was observed within the MHC region on chromosome 6. The association signal peaks at rs12189871 (odds ratio [OR]: 3.31, 95% confidence interval [CI] 3.21–3.40, $P = 1.8 \times 10^{-1524}$) but genome-wide significant evidence of association ($P < 5 \times 10^{-8}$) was observed from positions chr6:25,622,875 telomeric to and chr6:33,971,609 centromeric to the MHC, reflecting multiple independent associations and complex patterns of linkage disequilibrium (LD)[22] (Supplementary Fig. 1).

Genome-wide significant psoriasis associations were also observed in the present study at all but two of the previously reported susceptibility loci in Europeans[7–9] (Fig. 1, Supplementary Data 1, 'Methods'). Effect size estimates were consistent with the previous psoriasis meta-analysis[7] (Supplementary Fig. 2). The two loci without genome-wide significant evidence of association (13q14.11 and 21q22.12) were previously reported as psoriasis susceptibility loci in a trans-ethnic meta-analysis[23] and fell just short of genome-wide significance in the current study (Supplementary Data 2).

We identified associated genetic variants at 109 distinct loci, 50 of which have not been previously implicated in psoriasis susceptibility in European ancestry populations (Fig. 1, Supplementary Data 1 and 3, Supplementary Note). Four of these 50 loci encompass variants previously reported at genome-wide significance in other ancestral populations, either in Han Chinese studies (chromosomes 1p36.22, 2q11.2 and 4q27, albeit the reported lead variants are not associated in the current study)[24,25] or a trans-ancestry meta-analysis including

European and South Asian samples (1p36.22 and 1q24.2, with consistent effects in the current study)[22] (Supplementary Data 4). Effect size heterogeneity was observed at six of the 109 lead variants after accounting for multiple tests ($P_{het} < 4.6 \times 10^{-4}$), corresponding to relatively large effect loci (risk allele OR > 1.15) (Supplementary Fig. 3). A modest degree of heterogeneity is expected due to the range of ascertainment approaches across studies[26], with population-based biobank GWAS consistently exhibiting attenuated effect sizes (Supplementary Fig. 4).

To resolve the presence of multiple independent association signals at each of these loci we performed conditional and joint analysis[27] across 108 susceptibility loci (excluding the MHC). We found evidence of two or more high-confidence independent associations at 27 loci, including evidence of two independent signals each at newly reported loci at 4q27 and 8q12.2. In total we identified 148 independent non-MHC psoriasis association signals (Supplementary Data 1).

Assuming a prevalence of 1.5%, we estimate that 46.5% (standard error, SE: 4.4%) of variance in the liability to psoriasis is explained by common SNPs outside of the MHC region, increasing to 59.2% (SE: 13.2%) when including the MHC region. The 52 independent association signals at the 50 newly identified susceptibility loci are estimated to contribute 2.9% of the non-MHC common variant liability in addition to the 11.6% accounted for by established loci (Supplementary Data 5).

### Fine-mapping of candidate causal variants

Within each susceptibility locus, we sought to identify variants with strong statistical and functional evidence of being the causal variant underlying the psoriasis association signal. We constructed Bayesian 95% credible sets for causal variants at 144 sufficiently well-imputed independent susceptibility signals ('Methods'; Supplementary Data 1). A single variant had a posterior probability of >0.5 for being causal ($PP_{max} > 0.5$) at 51 (36%) of the fine-mapped non-MHC signals with resolution to a single putative causal variant ($PP_{max} > 0.95$) for 22 signals (15%) (Supplementary Data 6).

Our expanded GWAS meta-analysis provides increased power and improved ability to resolve causal variants. Thus, comparison against

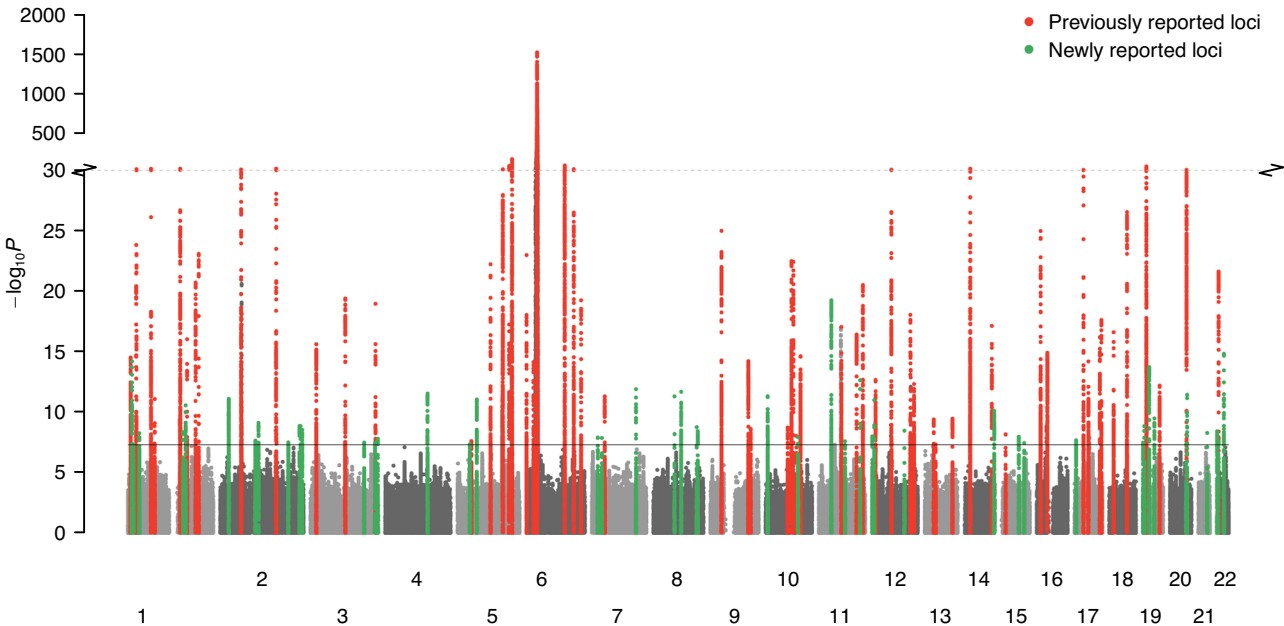

**Fig. 1 | Manhattan plot summarising genome-wide associations with psoriasis susceptibility.** x-axis, genomic position; y-axis, $-\log_{10}(P\text{-value})$ of association (two-sided Z-test, unadjusted for multiple tests); red and green points, regions previously and newly associated, respectively, with psoriasis susceptibility at genome-wide significance ($P = 5 \times 10^{-8}$) in European ancestry populations; solid horizontal line, genome-wide significance threshold; dotted horizontal line, y-axis break at $10^{-30}$; chromosomes (labelled 1–22) are alternately shaded for clarity.

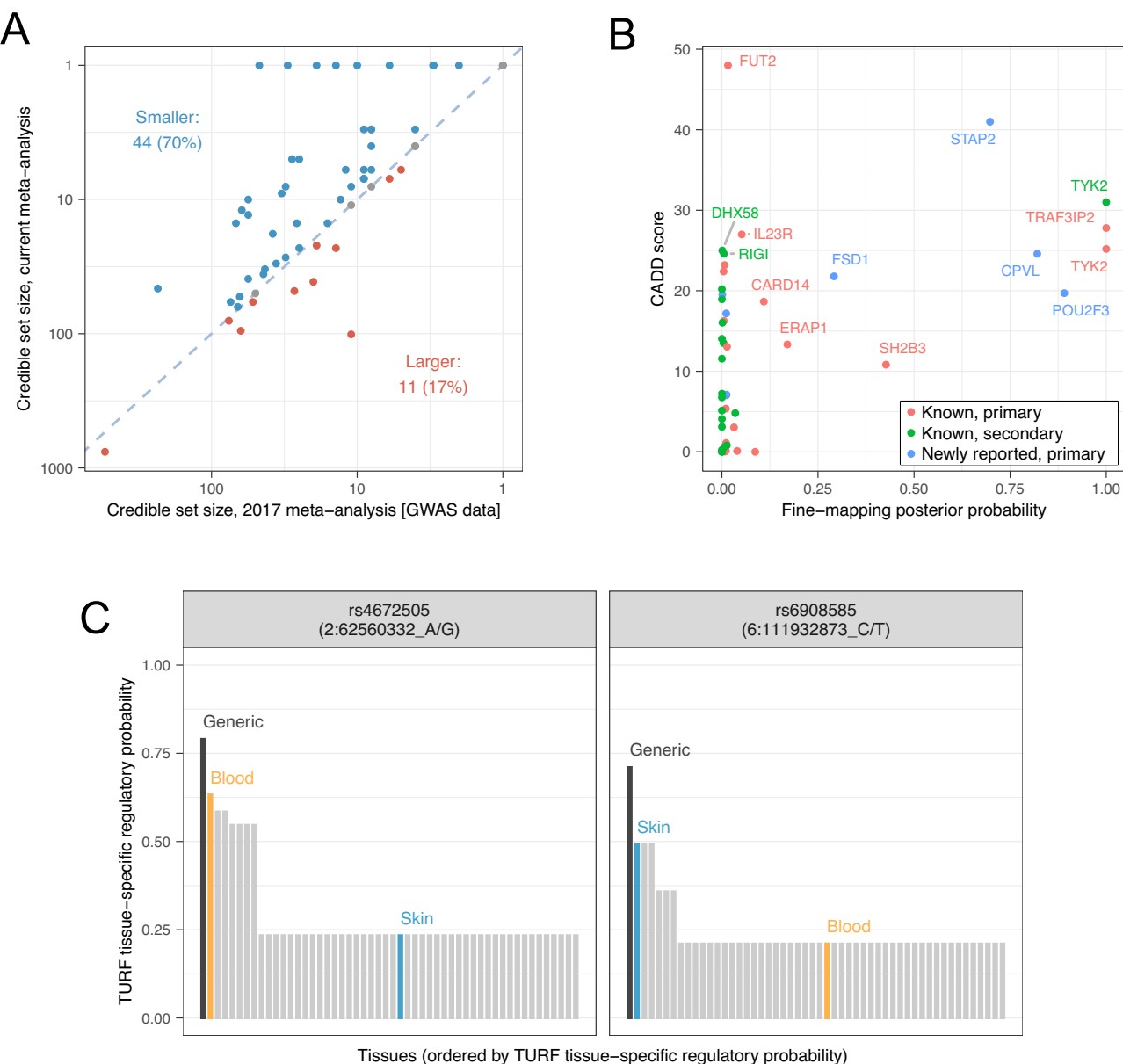

**Fig. 2 | Statistical and functional fine-mapping. A** Comparison of 95% Bayesian credible sets to previous GWAS meta-analysis. Each point represents a different association signal established in the previous meta-analysis (Tsoi et al., 2017). Point colour indicates direction of change, blue dashed line indicates equality. **B** Prioritisation of protein-altering variants. Points represent protein-altering variants identified in Bayesian credible sets for independent psoriasis signals; x-axis: posterior probability of causality from statistical fine-mapping analysis, y-axis: CADD score estimating deleteriousness of protein altering variant, point colour: whether corresponding susceptibility signal is in a known or newly reported genomic region and whether primary or secondary signal. Note the *TRAF3IP2* variant rs33980500 is discussed in the main text. **C** Highlighted high-confidence regulatory variants derived from TURF analysis. For each variant, bars show the generic and tissue-specific regulatory probabilities (y-axis) estimated by TURF for all tissues (x-axis). Blood and skin are highlighted in orange and blue, respectively.

fine-mapping in the previous meta-analysis[7] revealed the same or fewer number of variants in 95% credible sets at 52 (83%) of the 63 established psoriasis susceptibility loci (Fig. 2A, Supplementary Fig. 5, Supplementary Data 7).

To further highlight likely disease-causing variants and identify the biological mechanisms through which they influence pathological processes, we functionally annotated 5345 variants present in 95% credible sets. For 24 association signals, at least one variant in the credible set was predicted to give rise to altered or truncated protein sequences (Supplementary Data 8). Twenty of these missense and nonsense variants were predicted to be deleterious (CADD score > 15), including well-known alleles influencing psoriasis risk in *IL23R*, *TRAF3IP2*, *TYK2* and *RIGI* (formerly known as *DDX58*)[8]. Of relevance to *RIGI*, a deleterious missense

variant was also observed in the structurally related gene *DHX58* (ENSP00000251642.3:p.Asn461Ser, CADD score 25.0), encoding the nucleic acid receptor LGP2 which, like the protein retinoic acid-inducible gene (RIG-I) encoded by *RIGI*, contributes to the innate antiviral response. A further six putative causal variants were observed in genes located at novel psoriasis susceptibility loci including high-confidence deleterious variants affecting *CPVL* (ENSP00000387164.1:p.Tyr168His) and *POU2F3* (ENSP00000260264.4:p.His154Arg), and a low-frequency nonsense allele in *STAP2* that is protective against psoriasis (ENSP00000468927.1:p.Tyr169Ter, CADD score 41.0, OR = 0.79, 95% CI 0.76–0.82) (Fig. 2B). The fact that these candidate coding variants are identified in novel loci, at which moderate-to-large effect susceptibility variants are detected at lower minor allele frequencies than for known

loci (Supplementary Fig. 6), provides further justification that expanded psoriasis meta-analyses will continue to generate translational insights.

We also assessed the likelihood that variants within the credible sets have a regulatory function, both across tissues and specifically within blood or skin[28]. Consistent with the notion that statistical fine-mapping identifies causal variants, signals that resolve to smaller credible sets are enriched for variants with higher regulatory probability ($P_{generic} = 2.0 \times 10^{-11}$, $P_{tissue\text{-}specific} = 1.8 \times 10^{-17}$, Kruskal–Wallis test) (Supplementary Data 9, Supplementary Fig. 7). At 14 association signals, a single candidate variant was prioritised with a high generic regulatory probability, both in absolute terms and relative to other variants in the credible set (Supplementary Data 10, Supplementary Fig. 8). Notably, this included two independent signals at *TRAF3IP2*, a gene that encodes a regulator of NF-κB at chromosome 6q21[29–31]. We found strong evidence that the well-known psoriasis-associated missense variant rs33980500 (p.Asp10Asn; Fig. 2B)—which has been shown to impair binding to signalling molecules such as STAT3[32] and Hsp90[33]—may itself have a regulatory role, and independently that rs6908585 is a skin-specific regulatory variant (regulatory probability of 0.491; Fig. 2C). This suggests that both coding and gene regulation of *TRAF3IP2* contribute to psoriasis susceptibility at this locus.

## Identification of candidate genes

Given the demonstrated importance of putative regulatory variants at many of the established and newly identified psoriasis susceptibility loci, we estimated the degree to which the contribution to psoriasis risk of common genetic variation is mediated through effects on gene expression. We found 32.4% of heritability to be mediated by the *cis*-genetic component of gene expression levels across tissues, with between 13.7% and 18.3% in blood and skin (Supplementary Data 11), estimates that are broadly consistent with other complex disease traits[34].

Next we sought to identify genes whose expression is associated with psoriasis susceptibility variants in blood and skin (both sun exposed and unexposed) by means of a transcriptome-wide association study (TWAS) derived from Genotype-Tissue Expression (GTEx) project data[35]. As expected given the large effects of psoriasis MHC associations and the extensive LD characteristic of the region, and as characterised in detail previously[22], there are multiple genes within the MHC with predicted expression differences (Supplementary Fig. 9).

Outside of the MHC, and after filtering out low-confidence associations ('Methods'), transcriptome-wide significant gene expression differences ($P < 2.18 \times 10^{-6}$) were observed at 32 susceptibility loci (nine newly reported) and a further five loci within 1 Mb of association signals with suggestive evidence of association with psoriasis ($P_{meta} < 10^{-5}$) (Supplementary Data 12). Thirty-nine genes exhibited evidence in at least one tissue for a colocalized genetic association with psoriasis susceptibility and gene expression. Interestingly, the gene predicted to be differentially expressed was the closest gene to the lead variant of the association signal at only 12 (38%) of these 32 loci (Supplementary Fig. 10, Supplementary Data 13). Fourteen of the non-MHC TWAS genes have been previously highlighted as candidate psoriasis genes[10]. Amongst genes at new loci, *IRF5*, which encodes an interferon (IFN) regulatory factor that activates type I IFN responses, has been previously implicated in immune-mediated inflammatory disease through its association with systemic lupus erythematosus[36,37].

To identify groups of genes across psoriasis susceptibility loci with related biological function we employed DEPICT[38]. This highlighted a series of immune pathways and gene sets whose membership is over-represented at psoriasis risk loci (1632 gene sets at false discovery rate <5%; Supplementary Fig. 11, Supplementary Data 14A); this includes protein-protein interaction subnetworks for the genes *CBL* (proto-oncogene), *EGFR* (epidermal growth factor receptor; EGFR signalling being regulated via Cbl[39]), *ESX1* and *TEC* (Tec Protein Tyrosine Kinase) (Supplementary Data 14B) that are implicated for the first time in this study.

## Cellular and functional contexts of psoriasis associations

To identify the cellular contexts of genes whose transcription is mediated by psoriasis susceptibility variants, we asked which cells express TWAS target genes abundantly in the skin of psoriasis patients. We investigated expression patterns in single-cell transcriptomes derived from lesional and non-lesional skin of up to 14 chronic plaque psoriasis patients. Fifty-nine genes that were detectably expressed in scRNA-seq and identified in the TWAS analysis formed five clusters based on average expression patterns across twelve cell types (Fig. 3). The clusters exhibited prominent upregulation of genes in: (1) eccrine cells, (2) keratinocytes and melanocytes, (3) dendritic cells, (4) endothelial and lymphatic endothelial cells and fibroblasts, and (5) T cells, respectively. As expected, the genes located in the epidermal differentiation complex at chr1q21.3 were preferentially expressed in epidermal keratinocytes. Although not statistically significant after multiple testing, the genes assigned to each cluster demonstrated enrichment for cell type relevant functions, including immune response and regulation of autophagy (Supplementary Data 15).

In addition to the T cell centric regulation highlighted by previous studies, our results show that psoriasis-associated variants also govern gene regulation in stromal cells and keratinocytes. We therefore investigated whether specific cytokines regulate subsets of genes whose expression is influenced by psoriasis susceptibility variants, by examining the transcript abundance of non-MHC TWAS genes in transcriptome profiles from keratinocytes following a series of cytokine challenges (Supplementary Fig. 12, Supplementary Data 16)[40]. The transcriptomic shift most strongly enriched for TWAS genes was induced by IL36-alpha (observed/expected ratio = 3.8; $P = 3.0 \times 10^{-4}$), followed by a combination of IL17A and TNF (observed/expected ratio = 3.4; $P = 2.6 \times 10^{-3}$). These results suggest potential context-specific biological effects in keratinocytes for the implicated genes and highlight the roles of psoriasis loci in regulating inflammatory response in keratinocytes.

## Genome-wide correlations and causal inference

A series of recent studies have investigated the shared genetic architecture of psoriasis with other disease and health-related traits and have identified putative causal relationships with smoking, obesity and lifetime risk of cardiovascular disease[41–44]. To further investigate the shared genetic liability and potential causal relationships, we assessed genetic correlation between psoriasis and 345 disease and health-related traits and evaluated asymmetry in the correlation structure (specifically, the mixed fourth moments between effect sizes[45]) that are consistent with causal relationships.

Positive genetic correlations were observed between psoriasis and 36 diseases and health-related traits (Supplementary Data 17) including evidence of a substantial shared genetic architecture with colitis, generalised pain, angina and pulmonary disorders ($r_g \geq 0.30$, $P < 6.8 \times 10^{-6}$ in all cases; Supplementary Fig. 13). Significant genetic correlations were also observed with 42 physical and functional measures (Supplementary Fig. 14), and 11 traits related to lifestyle and quality of life (Supplementary Fig. 15).

Among traits for which we found evidence of a genetic correlation with psoriasis, 12 have evidence of asymmetry in the correlation structure that is consistent with a causal relationship (FDR < 0.05) (Supplementary Data 18, Supplementary Fig. 16). Susceptibility to stroke, triglyceride levels and multiple measures of adiposity have genetic support for a putative causal role in psoriasis (all genetic causal proportion [GCP] < −0.67, $P \leq 4.6 \times 10^{-3}$), the latter consistent with previous reports using Mendelian randomisation (MR)[41,42]. Notably, we also observed evidence indicating a causal role of psoriasis genetic risk in the development of other traits including back pain and generalised pain, fracture risk,

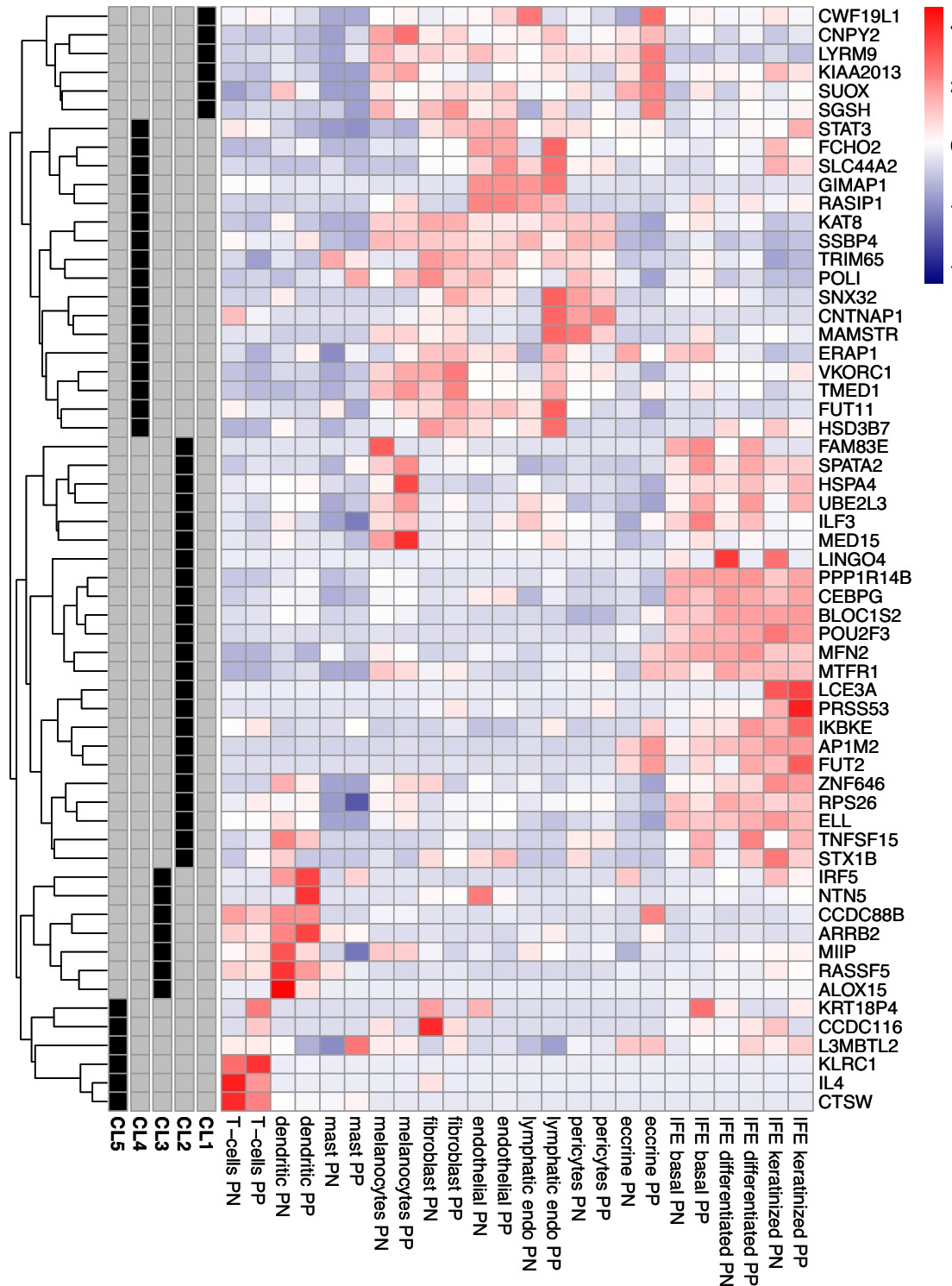

**Fig. 3 | Relative expression of TWAS genes in single-cell skin transcriptomes of psoriasis patients.** Expression level (cell colour; purple-red scale) represents mean value among cells in the corresponding cell type/condition (x-axis). For each gene (y-axis), the values were standardised and expression patterns were used for clustering (dendrogram, left hand side). Five clusters were identified (CL1-CL5, grey/black bars, left hand side) and labelled according to the cell types that exhibit highest expression for the genes in the cluster. PP, psoriasis lesions; PN, non-lesional skin; Lymphatic Endo, lymphatic endothelial cells; IFE, interfollicular epidermis. Clusters are identified based on enrichment for: (1) eccrine cells; (2) keratinocytes and melanocytes; (3) dendritic cells; (4) endothelial and lymphatic endothelial cells and fibroblasts; (5) T cells.

diabetes and periodontitis (all GCP > 0.63, $P < 3.2 \times 10^{-3}$). Five of the 12 causal relationships were observable using two-sample MR (Supplementary Data 19), the other seven highlighting the additional value of considering genome-wide variation through the GCP-based approach.

## Discussion

Relative to previous genetic studies of psoriasis susceptibility[7], the increased statistical power of our larger meta-analysis sample size has resulted in the identification of 46 new psoriasis susceptibility loci with genome-wide significant evidence of association. A further four risk

loci have been observed for the first time at genome-wide significance in populations of European ancestry.

A range of recent and emerging targeted therapies have proven highly effective in psoriasis[1]. Many of these exemplify the concordance between a drug's efficacy and the presence of disease susceptibility variants that disrupt the targeted pathway[17], including biologics targeting IL-23 (encoded by *IL23A* and *IL12B* at the 12q13.3 and 5q33.3 psoriasis susceptibility loci, respectively) and deucravacitinib[46] a small-molecule TYK2 inhibitor whose development was directly informed by the presence of loss-of-function alleles of *TYK2* that confer protection to psoriasis[8]. Furthermore, the efficacy of TNF inhibitors has been associated with genetic variants at the 6q23.3 susceptibility locus, where *TNFAIP3* encodes A20, an inhibitor of TNF signalling[47]. The current study establishes two new psoriasis susceptibility loci harbouring genes encoding therapeutic targets (Table 1, Supplementary Data 20). We found novel psoriasis susceptibility variants centred on the 5′ untranslated region of *IL17RA* at chromosome 22q11.1 (lead variant rs917864, OR: 1.08, $P = 3.9 \times 10^{-9}$). The interleukin-17 receptor A subunit encoded by this gene[48] is bound with high affinity by brodalumab, a biologic therapy demonstrated to confer effective long-term control of psoriasis[49,50]. Similarly, the aryl hydrocarbon receptor is encoded by *AHR*, the closest protein-coding gene to the novel 7p21.1 association signal (lead variant rs78233367, OR: 1.06, $P = 1.3 \times 10^{-8}$). Activation of this receptor by the recently approved topical AHR-modulating agent Tapinarof effectively controls psoriasis symptoms[51–53].

Our results also have implications for non-pharmacological interventions to manage psoriasis. We present evidence that increased adiposity, measured by body mass index, fat mass or waist circumference, leads to elevated psoriasis risk. This finding is consistent with previous observational studies demonstrating that psoriasis incidence and severity in obese individuals can be reduced through adiposity-reducing lifestyle interventions[54], and with observed improvements in comorbid psoriasis in type 2 diabetes patients treated with weight loss-inducing glucagon-like peptide 1 (GLP-1) receptor agonists[55]. Both adiposity and metabolic profile, the latter congruent with the causal role of triglycerides that our study highlights, may mediate the hypothesized effects of diet on psoriasis severity that are the subject of several ongoing trials[56,57].

In addition to AHR, a series of other transcription factors were implicated as putative causal genes by our functional fine mapping approach (Table 1, Supplementary Data 20). We observed examples of newly reported association signals mapping to fundamental processes of cell differentiation, proliferation and trafficking that contribute to established psoriasis pathomechanisms including interferon response, T cell regulation and keratinocyte hyperproliferation. The elongation factor gene *ELL*, which maps to the newly identified 19p13.11 locus, is predicted by TWAS to be upregulated in blood in the presence of psoriasis-associated alleles. Moreover, by stabilizing RNA polymerase II within the super elongation complex, ELL has been demonstrated to sustain the expression of the epidermal proliferation genes that are greatly upregulated in psoriasis[58]. A deleterious missense variant in *POU2F3* (rs7110845, p.His154Arg, $P = 2.3 \times 10^{-13}$), which encodes an epidermal transcription factor critical to keratinocyte differentiation[59], is reported here for the first time. CCAAT enhancer binding transcription factors, implicated in myeloid lineage commitment and regulation of inflammatory cytokines[60–62], are encoded by *CEBPB* at the known 20q13.13 locus, and *CEBPA* and *CEBPG* at the newly reported 19q13.11 locus. The latter gene is predicted by TWAS to be downregulated in sun-exposed and unexposed skin in the presence of psoriasis risk variants and would therefore be consistent with previous reports that C/EBPγ suppresses proinflammatory cytokine activity[63].

Transcriptional regulation of interferon responses is further coordinated by genes at a series of psoriasis susceptibility loci. The interferon regulatory factor gene *IRF5* is predicted by TWAS to be upregulated across tissues in the presence of psoriasis risk variants at the newly reported 7q32.1 locus. We also identified a susceptibility locus at 20q13.2 that contains *NFATC2*, encoding a transcription factor that interacts with interferon regulatory factor 4 (encoded by *IRF4* at the 6p25.3 locus) to regulate T cell development via enhanced IL-4 expression. Epigenetic regulation of interferon signalling is implicated through the histone acetyltransferase gene *KAT8* (16p11.2), which directly regulates the transcriptional activity of IRF3[64], and is predicted to be upregulated in skin.

We shed further light on the cellular context of the interferon response in psoriasis through our single-cell expression data. Consistent with its role in in epidermal proliferation[58], we found that *ELL* is a TWAS candidate predominantly expressed in keratinocytes (Fig. 3). In the same cluster, another group of TWAS target genes are highly expressed in melanocytes from psoriasis lesions. In addition to producing melanin for photoprotection, melanocytes assume an activated phenotype in psoriasis lesions[65] and express functional toll-like receptors TLR2, 3, 4, 7 and 9[66], enabling innate antibacterial and antiviral immune responses. Melanocytes have also been recently proposed as target cells of the HLA-C*06:02-restricted autoimmune response in psoriasis via the melanocyte autoantigen ADAMTSL5, which could provide a skin-specific target for autoimmune attack[67,68]. More generally, our single-cell expression analysis suggests that psoriasis genetic risk is mediated by a range of cell types beyond the established triad of T cells, dendritic cells and keratinocytes.

We found predicted deleterious protective variants in two genes encoding members of the RIG-I like receptor (RLR) family, linking recognition of viral RNAs to interferon signalling[69,70]: *DHX58* at chromosome 17q21.2 and *RIGI* (formerly *DDX58*) at 9p21.1 (Supplementary Data 8). Prior to this study, *STAT3* had been considered the leading functional candidate in 17q21.2 due to its prominent role in IL-23 signal transduction[10]. Previous reports have implicated deleterious coding variants in *IFIH1*, the third member of the RLR family, as conferring protection against psoriasis[8,71] and psoriatic arthritis[72]; the protective *IFIH1* variants rs35667974 and rs1990760 are both highly significant in the current meta-analysis ($P = 5.0 \times 10^{-23}$ and $7.0 \times 10^{-50}$, respectively) but not prioritised in credible sets due to insufficient effective sample size and the presence of a more strongly associated intronic variant (rs2111485), respectively. While the *RIGI* and *DHX58* coding variants both show a low posterior probability of being causal (the intronic variant rs11795343 being strongly preferred in *RIGI*, with posterior probability 0.922), the fact that potentially deleterious coding variants are found to be associated with reduced psoriasis risk in all three known human RLR family members is intriguing. These findings are consistent with reports of rare gain-of-function mutations in *RIGI* and *IFIH1* underlying Singleton-Merton syndrome, in which psoriasis is a clinical feature[73,74]. Much remains to be learned about how the role of RLR family members in driving interferon signalling is affected by psoriasis-associated genetic variation.

We note that differing ascertainment strategies among the constituent studies, with consequent variation in clinical presentation and severity of sampled psoriasis cases, likely contribute to the effect size heterogeneity observed at six loci and slightly attenuated effect size estimates at previously established European loci (Supplementary Fig. 2). Furthermore, 24 susceptibility loci reported in non-European populations were not observed with genome-wide significant evidence of association (Supplementary Data 21, Supplementary Fig. 17). Inter-population differences in allele frequency or local LD structure may partially account for this observation, with 11 (46%) of the lead variants less than 500 kb from a variant with suggestive evidence of association ($P < 1.0 \times 10^{-5}$) in the current study. The remaining 13 (54%) may represent ancestry-specific associations, and further studies are needed to address the important question of how genetic variation influences psoriasis risk across ancestry groups[75,76].

**Table 1 | Candidate causal genes at newly reported psoriasis susceptibility loci**

| Chr | LD block | Lead psoriasis susceptibility variant | EA/NEA | Odds ratio (95% CI) | P-value | Genes with protein-altering variants | TWAS candidates | Targets for existing psoriasis therapies |
|---|---|---|---|---|---|---|---|---|
| 1 | 11710076–12266177 | rs78088488 | CA/C | 0.919 (0.899, 0.939) | $6.3 \times 10^{-15}$ | – | KIAA2013, MFN2, MIIP | – |
| 1 | 15686534–17547392 | rs4481843 | A/G | 0.943 (0.924, 0.963) | $4.6 \times 10^{-8}$ | – | RP11-169K16.8 | – |
| 1 | 31218841–34056003 | rs34190690 | T/C | 1.068 (1.045, 1.092) | $5.6 \times 10^{-9}$ | – | ADC | – |
| 2 | 107780988–112030011 | rs13029175 | T/G | 0.942 (0.924, 0.96) | $7.7 \times 10^{-10}$ | ACOXL | – | – |
| 5 | 34566074–36433954 | rs4594881 | T/G | 0.949 (0.931, 0.967) | $4.7 \times 10^{-8}$ | IL7R | – | – |
| 7 | 16654963–17593990 | rs78233367 | T/C | 1.063 (1.041, 1.086) | $1.3 \times 10^{-8}$ | – | – | AHR (e.g. by tapinarof) |
| 7 | 28712513–29197970 | rs117744081 | A/G | 0.867 (0.825, 0.911) | $1.4 \times 10^{-8}$ | CPVL | – | – |
| 7 | 127908745–129931330 | rs4728141 | T/C | 0.937 (0.92, 0.954) | $1.3 \times 10^{-12}$ | – | IRF5 | – |
| 11 | 117298216–120390709 | rs7110845 | A/G | 1.07 (1.051, 1.09) | $2.3 \times 10^{-13}$ | POU2F3 | POU2F3 | – |
| 17 | 4455699–5737306 | rs139021596 | CACAA/C | 0.93 (0.907, 0.954) | $2.1 \times 10^{-8}$ | PELP1 | ALOX15, ARRB2 | – |
| 19 | 3906651–4551554 | rs79657645 | C/G | 0.79 (0.732, 0.853) | $1.4 \times 10^{-9}$ | FSD1, STAP2 | – | – |
| 19 | 17783937–19065462 | rs8107351 | A/G | 1.075 (1.055, 1.095) | $1.9 \times 10^{-14}$ | – | SSBP4, ELL | PDE4C (e.g. by apremilast)[a] |
| 19 | 32573487–34140146 | rs75826800 | T/C | 0.935 (0.916, 0.955) | $3.3 \times 10^{-10}$ | PEPD | CEBPG | – |
| 22 | 1–17713187 | rs917864 | T/C | 1.075 (1.05, 1.101) | $3.9 \times 10^{-9}$ | – | – | IL17RA (e.g. by brodalumab) |
| 22 | 19717883–20948848 | rs165687 | A/G | 1.062 (1.04, 1.085) | $2.4 \times 10^{-8}$ | – | MED15 | – |

Full details of lead variant association statistics are presented in Supplementary Data 1.
Chr chromosome, LD linkage disequilibrium, EA effect allele, NEA non-effect allele, CI confidence interval, P-value meta-analysis association p-value (two-sided Z-test, unadjusted for multiple tests).
[a]PDE4A is located at the known psoriasis susceptibility locus on chromosome 19p13.2.

Finally, our TWAS analysis relies on reference data from healthy participants to link susceptibility-associated genetic variation to predicted expression differences in skin and blood. TWAS have demonstrated utility in prioritizing causal genes but nonetheless harbour the potential for false positive findings when applied to incorrect tissues[77]. While we provide additional context for our TWAS candidate genes in psoriatic single-cell expression data and in cytokine-stimulated keratinocytes, future studies are needed to map causal relationships from psoriasis risk alleles to variation in the transcriptomic profiles of psoriasis patients undergoing systemic inflammation. These efforts will benefit both from larger and higher resolution context-specific functional genomic datasets, encompassing a wider range of skin and blood cell types, and from emerging algorithms that can integrate these data with aligned genetic associations to implicate specific cell types and their contexts in the psoriasis disease process[78,79].

This study represents a major advance in our understanding of the genetic basis of psoriasis. The number of documented psoriasis susceptibility signals has approximately doubled, with better refinement of known loci to highlight plausible biological mechanisms through which they influence psoriasis risk. We propose novel disease mechanisms, including a role for the disruption of basic cellular machinery such as transcription and epigenetic modulation in regulating the inflammatory process in psoriasis. For the first time, genetic susceptibility signals are contextualised to specific skin cell types and cytokine signalling pathways. Our data point to the participation in psoriasis pathogenesis of previously underappreciated cell types such as melanocytes. This work will underpin the next era of molecular studies in psoriasis and psoriasis therapeutics.

## Methods

### Contributing GWAS studies

Genotype data for chronic plaque psoriasis cases and unaffected or population-based controls were compiled for 18 contributing studies, each of which underwent stringent QC at one of eight contributing analysis centres (Supplementary Data 22). Full details are provided in Supplementary Data 23. All studies obtained local ethical approval and complied with relevant ethical regulations, and participants provided informed consent (Supplementary Data 23). Prior to association testing, inter-dataset duplicated and first- or second-degree related participants were identified using KING (version 2.0)[80] by sharing subsets of between 2502 and 6864 genotyped markers outside known psoriasis-associated regions (see Supplementary Methods for justification of these marker counts). After removing duplicated and related samples, the final sample size was 36,466 cases and 458,078 controls. The cumulative effective sample size was 103,614 (Supplementary Data 22). All participants were of European ancestry and sex-stratified analyses were not performed in this study. The contributing analysis centres were responsible for genome-wide imputation and association testing (further details in Supplementary Methods), and provided effect size estimates for each tested variant (i.e., regression beta coefficients [log-odds ratios] and associated standard errors) to be meta-analysed.

### Meta-analysis

Summary statistics from individual studies were aligned based on GRCh37 positions and alleles; reference alleles were checked for consistency across studies. In many datasets, more than one imputation reference panel was used to maximise imputation quality[9]; summary statistics for each variant were preferentially taken from the version with highest imputation quality (INFO or $R^2$) score. Standard error-weighted fixed effects meta-analysis was performed using METAL v2020-05-05[81].

Genomic inflation was calculated based on a set of 170,786 LD-independent variants outside of previously established psoriasis susceptibility loci having minor allele frequency >0.05 (Supplementary

Methods)[20]. LD score regression intercept and the proportion of inflation attributed to causes other than polygenic heritability (LDSC ratio) were calculated using LDSC v1.0.1 software with default settings and precomputed LD scores derived from 1000 Genomes data[21].

## Definition of susceptibility loci and independent signal identification

Psoriasis susceptibility loci were identified using Genome-wide Complex Trait Analysis conditional and joint analysis (GCTA-COJO)[27]. To facilitate this, the autosomes were first partitioned into distinct LD blocks (Supplementary Methods). A region (LD block) was considered to be a psoriasis susceptibility locus if at least one variant tested in three or more contributing studies and with $N_{eff} > 10,000$ (11,808,957 eligible variants in total) achieved genome-wide significance ($P < 5 \times 10^{-8}$). The LD block that includes the MHC region (chr6, 24.0–36.3 Mb) extends beyond the established boundaries of the extended MHC (chr6, 25.7–33.4 Mb[82]); we observe an association outside of the established boundaries that could plausibly be driven by weak LD with the strongly associated MHC variants (Supplementary Fig. 1).

For the identification of additional independent association signals within associated regions we used a more stringent subset of variants having $N_{eff} > 93,252$ (90% of maximum possible), with five non-MHC regions omitted due to lack of variants meeting this sample size threshold. We employed GCTA-COJO with a custom reference panel to determine independently associated lead variants using a stepwise model selection procedure, and to estimate conditional association statistics for each signal in LD blocks with multiple independent signals (Supplementary Methods). We were unable to estimate additional independent signals within the MHC block (Supplementary Methods).

To annotate our genomic regions against psoriasis susceptibility loci previously established at genome-wide significance in European populations or other ancestries, we reviewed recent GWAS meta-analyses[7–9,22–25,83] and our recent review[10], and assessed all psoriasis associations in GWAS Catalog (accessed 31 October 2022)[84]. We considered previously reported associations to be recapitulated if they fell within the same LD block as a genome-wide significant signal (i.e. lead variant) in the current study. Note that association signals previously reported as distinct loci but falling within the same LD block are classed as a single locus in the current study. This occurred at four (current study) loci, albeit in each case multiple independent signals were identified (Supplementary Data 1). Effect size consistency against the previous (2017) meta-analysis[7] was assessed based on the lead variants at previous meta-analysis loci, using marginal effect estimates and without regard to their correspondence to primary or secondary association signals within newly defined loci.

We used LDSC[21] to estimate the total common SNP heritability for psoriasis on the liability scale, assuming a population prevalence of 1.5%. We note that the heritability estimated by LDSC when including the MHC region is to be interpreted with caution due to its complex genetic architecture[21,85]. Variance explained by our genome-wide significant associations was estimated using the Mangrove (v1.21) package in R[86], based on the jointly estimated effect sizes from COJO and with allele frequencies estimated from 1000 Genomes data. Since independent signals were not estimated in the LD block containing the MHC region, only the marginal effect size of the lead SNP was included, which is likely to underestimate the heritability attributable to this region.

## Statistical fine-mapping

For each independent association signal, we used the method suggested by Wen and Stephens[87], and implemented (including the specification of prior distributions) in previous work[22], to calculate posterior probabilities (PP) for each variant being causal. Within each LD block, PPs were calculated for variants with $N_{eff} > 90\%$ of maximum, using the joint meta-analysis association $p$-values estimated by GCTA-

COJO to account for the presence of multiple association signals. Bayesian 95% credible sets were subsequently constructed by incorporating variants in decreasing PP order until the cumulative PP exceeded 0.95. Since independent signals could not be established for the LD block containing the MHC region, the 95% credible set for the lead signal was based on unconditional association statistics. We did not estimate credible sets for five signals where no variants had $N_{eff} > 90\%$ of maximum.

We also assessed improvement in statistical fine-mapping relative to the GWAS datasets available in the previous (2017) psoriasis GWAS meta-analysis[7]. To this end, we selected 63 psoriasis susceptibility loci that were genome-wide statistically significant in either the 2017 meta-analysis and/or other published studies of white European ancestry populations and, for comparability, re-computed associations for these loci in the 2017 meta-analysis without the PAGE dataset (which was typed using the Immunochip and therefore lacks full genome-wide coverage). For the comparison, 95% credible sets were calculated in both the current and (re-computed) 2017 meta-analysis within windows of 200 kb around the 2017 lead markers, based on unconditional association statistics; credible set construction was restricted to variants well-imputed for a $N_{eff} > 90\%$ of the maximum possible meta-analysis sample size.

## Prioritisation of causal variants

For variants identified in 95% credible sets (5344 distinct variants), we assessed the potential contribution to disease risk via altered protein-coding or regulation. Variants with predicted protein-altering consequence were identified using the Variant Effect Predictor[88], along with associated Combined Annotation Dependent Depletion (CADD) scores[89]. To identify regulatory variants likely to influence psoriasis risk we calculated generic and tissue-specific regulatory prediction scores with the RegulomeDB-based method TURF[28]. The TURF analyses used default settings and reference data, and excluded indels (5102 SNPs in total) A (one-sided) Kruskal-Wallis test was used to determine whether variant TURF scores were identically distributed for different ranges of credible set sizes. For each independent association signal, we checked for prioritised regulatory variants, which we defined as variants with: (i) generic regulatory probability >0.7, (ii) at least 50% share of generic regulatory probability among variants within their credible set, and (iii) no other credible set variant having generic regulatory probability >0.5. We assessed blood- and skin-specific regulatory probabilities for prioritised variants and compared these to 49 other tissues with tissue-specific regulatory probabilities.

## Transcriptome-wide association study

The proportion of psoriasis heritability mediated by genetic effects on gene expression was estimated using mediated expression score regression (MESC)[34]. We estimated mediated heritability using pre-computed expression scores meta-analysed across all tissues, and specific to whole blood, sun-exposed skin and unexposed skin, in GTEx v8 (https://github.com/douglasyao/mesc/wiki/Download-expression-scores), with stratified LD scores (baseline v2.0) derived from 1000 Genomes Phase 3 reference data[85].

Predictions of differential gene expression in the presence of psoriasis-associated genetic variation were generated using the S-PrediXcan method[90] implemented in the Complex Trait Genetics Virtual Lab (CTG-VL)[91]. We predicted expression differences in whole blood ($n = 6275$ genes with prediction models available), sun-exposed skin ($n = 9085$) and sun-unexposed skin ($n = 7617$) using GTEx v7 reference data[92]. We employed a Bonferroni-adjusted transcriptome-wide significance threshold of $2.18 \times 10^{-6}$ (22,977 tests performed) to assess differential expression associated with psoriasis. As recommended by the authors of S-PrediXcan[90], we performed post-hoc filtering of the results to identify high-confidence TWAS associations, removing genes with non-significant prediction performance or

evidence of non-colocalizing psoriasis and expression signals (further details in Supplementary Methods).

To establish the physical location of psoriasis-associated TWAS genes relative to susceptibility signals, we considered all protein-coding and long non-coding RNA genes from Ensembl (GRCh37 version 109)[93]. Gene positions were annotated using the biomaRt (v2.42.1) R package[94]. Genes were allocated to psoriasis-associated genomic regions based on having transcription start site (TSS) within the corresponding LD block, or within 1 Mb of the lead variant. Gene distances in each region were compared based on the distance from TSS to the lead psoriasis-associated variant. Significant TWAS genes outside of psoriasis-associated genomic regions were mapped based on distance to the variant within 1 Mb having lowest meta-analysis p-value (all psoriasis-suggestive variants, $P_{meta} < 10^{-5}$).

### Single-cell functional analyses

To delineate the specific skin cell types underlying psoriasis-associated TWAS genes, we studied their expression profiles using skin-derived scRNA-seq data that we recently reported[95]. For each TWAS gene, we computed the average expression in each of 12 cell types derived from non-lesional or lesional skin of psoriasis cases (24 biopsy type/cell type combinations in total; $n_{non-lesional} = 11$, $n_{lesional} = 14$), and subsequently calculated the standardized expression across cell types for each gene. We grouped genes with consistent expression profiles and identified five clusters using hierarchical clustering.

For each cluster, functional enrichment testing was performed using a one-sided hypergeometric test for overrepresentation in 12,729 Gene Ontology Biological Process[96] and Human Phenotype Ontology[97] gene-sets derived from the Molecular Signatures Database[98]; false discovery rate was calculated using the Benjamini-Hochberg procedure.

We also investigated the overlap between psoriasis-associated TWAS genes and transcriptomic changes induced by psoriasis-linked cytokines. Specifically, we utilised publicly available gene expression profiles generated for keratinocytes from 50 healthy adult donors before and after stimulation with a range of cytokines (IL-4, IL-13, IFN-α, IFN-γ, TNF-α, and IL-17A)[40,99]. We compared and tested for statistical enrichment (one-sided hypergeometric test) our list of non-MHC TWAS genes against the statistically significant upregulated genes (fold change ≥1.5, FDR ≤ 10%) in each keratinocyte cytokine challenge.

### Biological pathway analysis

We used DEPICT v1.1 to identify sets of genes with coordinated function in which genes from psoriasis susceptibility loci are highly expressed[38]. As input we used a list of 214 independent and genome-wide significant variants derived from the full meta-analysis summary statistics via distance- and LD-based clumping ($r^2 > 0.05$ within 2 Mb windows) in PLINK[100] based on 1000 Genomes European samples[101], except for the extended HLA region where only the lead SNP was included. For comparison purposes, we repeated the analysis using a subset of 158 variants within 1 Mb of previously established psoriasis susceptibility loci. In each case, functional gene sets were considered significantly enriched at a false discovery rate (FDR) of <0.05 based on 14,462 gene sets tested.

### Correlation and causation analyses

Genetic correlations[102] were estimated in CTG-VL using summary statistics for an initial set of 1376 traits (CTG-VL default list), a majority of which were derived from UK Biobank by the Neale lab (http://www.nealelab.is/uk-biobank). We excluded traits that were too general or non-specific, represented follow-up questions for specific groups (e.g., details of presentation for severe mental illnesses) or detailed lifestyle questions (e.g., specifics of diet or occupation), psoriasis and duplicated or otherwise difficult-to-interpret traits, resulting in 592 traits categorised into disease and health ($n = 345$), physical and functional measures ($n = 210$) or lifestyle and quality of life ($n = 37$) (Supplementary Data 17). Significant genetic correlation was identified among these traits based on a Bonferroni-adjusted $p$-value threshold of $8.45 \times 10^{-5}$ (592 tests performed), and we checked that all significant traits had heritability z-score >4.

Partial genetic causality between psoriasis and a range of other traits was estimated using a latent causal variable (LCV) model[45] implemented in CTG-VL. Of the 592 traits for which genetic correlation was assessed (above), LCV results were available for 585. We focused on 108 traits with significant genetic correlation at FDR < 0.05. Traits with evidence for a causal relationship were subsequently identified based on a significant genetic causality proportion (FDR < 0.05 among 108 traits). The twelve identified causal relationships were also assessed bidirectionally using two-sample Mendelian randomization (Supplementary Methods).

### Reporting summary

Further information on research design is available in the Nature Portfolio Reporting Summary linked to this article.

## Data availability

The meta-analysis summary statistics generated in this study have been deposited in the GWAS Catalog under accession code GCST90472771. This study used a custom LD reference panel comprising six GWAS datasets. Individual level genotype data for the CASP GWAS, PsA GWAS, and Exomechip case-control studies are available on dbGaP (dbGaP: phs000019.v1.p1, phs000982.v1.p1, and phs001306.v1.p1, and WTCCC2 genotype data are archived at the European Genome-Phenome Archive (study ID EGAS00000000108). Data sharing restrictions do not allow making genotype data publicly available for the remaining two case-control cohorts. However, LD matrices based on the full reference panel for all 109 susceptibility loci have been deposited in the King's College London research data repository, KORDS, at https://doi.org/10.18742/27982057. This study used publicly available reference data accessed through bioinformatics tools and provided by their developers: LD scores with LDSC, variant annotations with VEP and TURF, expression scores with MESC, gene annotations with DEPICT. This study used GTEX v7 eQTL data, accessed at https://gtexportal.org/home/downloads/adult-gtex/qtl. This study used annotated gene sets from the Molecular Signatures Database v2023.2.Hs (https://www.gsea-msigdb.org/gsea/msigdb/). The scRNA-seq data used in this study are deposited in the Gene Expression Omnibus under accession code GSE173706, and the cytokine-stimulated keratinocyte expression data under accession code GSE255828. Genetic correlation and causal analysis were conducted for a wide range of traits using GWAS summary statistics compiled by the Complex Trait Genetics Virtual Lab (https://vl.genoma.io/). Further analyses were conducted using GWAS summary statistics made available by the Neale Lab (http://www.nealelab.is/uk-biobank/), the GLIDE consortium (https://data.bris.ac.uk/data/dataset/2j2rqgzedxlq02oqbb4vmycnc2), or deposited in GWAS Catalog under accession code GCST002216.

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

## Acknowledgements

We gratefully acknowledge the contributions of patients and family members, other research participants and clinical staff involved in recruitment. N.D. is funded by Health Data Research UK (MR/S003126/1). Support for the study was received from the Department of Health through the National Institute for Health and Care Research (NIHR) Bio-Resource Clinical Research Facility and comprehensive Biomedical Research Centre awards to Guy's and St Thomas' National Health Service Foundation Trust in partnership with King's College London and King's College Hospital National Health Service Foundation Trust (BRC_1215_20006). We acknowledge support from the Psoriasis Association in relation to Biomarkers of Systemic Treatment Outcomes in Psoriasis (RG2/10, RG2/10) and an award to P.D.M. (ST1/19) and FC (ST3/20). This project has received funding from the Innovative Medicines Initiative 2 Joint Undertaking (JU) under grant agreement number 821511 (Biomarkers in Atopic Dermatitis and Psoriasis). The JU receives support from the European Union's Horizon 2020 research and innovation programme and the European Federation of Pharmaceutical Industries and Associations (EFPIA). This publication reflects only the authors' views and the JU is not responsible for any use that may be made of the information it contains. S.K.M. is funded by a NIHR Advanced Fellowship (NIHR302258). This study presents independent research supported by NIHR BioResource Centre Maudsley, NIHR Maudsley Biomedical Research Centre (BRC) at South London and Maudsley NHS Foundation Trust and Institute of Psychiatry, Psychology and Neuroscience (IoPPN), King's College London. The BioResource gratefully acknowledge capital equipment funding from the Maudsley Charity (Grant Ref. 980) and Guy's and St Thomas's Charity (STR130505). P.D.M. reports a Translational Research Grant (814364) from the National Psoriasis Foundation. L.C.T. received funding from the National Institutes of Health (NIH) (K01 AR072129, P30 AR075043, UC2 AR081033). J.T.E. was supported by awards from NIH (R01AR042742, R01AR050511, R01AR054966, R01AR063611, R01AR065183), the National Psoriasis Foundation (NPF) including an NPF Bridge Grant, the Babcock Memorial Trust, and the Ann Arbor Veterans Affairs Hospital. M.T.P. is supported by a Research Career Development Award from the Dermatology Foundation. X.W. is supported by NIH (R01ES033634, R35GM138121). M.K.S. reports a Translational Research Grant from the NPF. A.C.B. was supported by the University of Michigan A. Alfred Taubman Medical Research Institute via Taubman Emerging Scholar funds and the NIH (K08 AR078251, P30 AR075043). J.E.G. is supported by NIH (P30AR075043) and the Taubman Medical Research Institute. Part of this study was funded by a grant to A.R. and U.H. from the Bundesministerium für Bildung und Forschung (BMBF Metarthros 01EC1407A), by a grant to U.H. from the German Research Foundation (CRC1181-2, project A05) and by a grant to H.B. and F.B. from the Bundesministerium für Bildung und Forschung (BMBF ArthroMark 01EC1401C). The HNR study (Erlangen cohort) is supported by the Heinz Nixdorf Foundation (Germany). Additionally, the study is funded by the German Ministry of Education and Science and the German Research Council (DFG; Project SI 236/8-1, SI236/9-1, ER 155/6-1). The genotyping of the Illumina HumanOmni-1 Quad BeadChips of the HNR subjects was financed by the German Centre for Neurodegenerative Disorders (DZNE), Bonn. T. Traks and K.K. report support from the Estonian Research Council (PUT1465, PRG1189). S.K. is supported by MSWA, The Michael J. Fox Foundation, Shake It Up Australia, and Perron Institute for Neurological and Translational Science. M.T.-L. was supported by the European Union through the European Regional Development Fund (Project No. 2014-2020.4.01.15-0012). R.M. was supported by Estonian Research Council grant PRG1911. T.E. was supported by Estonian Research Council grant PRG1291. The Trøndelag Health Study (The HUNT Study) is a collaboration between HUNT Research Center (Faculty of Medicine and Health Sciences, NTNU - Norwegian University of Science and Technology), Trøndelag County Council, Central Norway Regional Health Authority, and the Norwegian Institute of Public Health. The genotyping in HUNT was financed by the National Institutes of Health; University of Michigan; the Research Council of Norway; the Liaison Committee for Education, Research, and Innovation in Central Norway; and the Joint Research Committee between St Olavs hospital and the Faculty of Medicine and Health Sciences, NTNU. L.T., K.H. and M.L. work in a research unit funded by Stiftelsen Kristian Gerhard Jebsen; Faculty of Medicine and Health Sciences, NTNU - Norwegian University of Science and Technology; The Liaison Committee for Education, Research and Innovation in Central Norway; the Joint Research Committee between St. Olavs Hospital (Trondheim, Norway) and the Faculty of Medicine and Health Sciences, NTNU - Norwegian University of Science and Technology. M.L. is supported by grants from the Liaison Committee for Education, Research and Innovation in Central Norway and the Joint Research Committee between St Olav's hospital and the Faculty of Medicine and Health Sciences, NTNU. The study received support from the Deutsche Forschungsgemeinschaft (DFG, German Research Foundation) Cluster of Excellence 2167 Precision Medicine in Chronic Inflammation (PMI) (EXC 2167-390884018). This work was supported by a grant from Versus Arthritis (21754). This research was co-funded by the NIHR Manchester Biomedical Research Centre (NIHR203308). The views expressed are those of the author(s) and not necessarily those of the NHS, the NIHR or the Department of Health. D.J. acknowledges that his research was supported by Cambridge Arthritis Research Endeavour (CARE) and the NIHR Cambridge Biomedical Research Centre (BRC-1215-20014). W.L. acknowledges funding from NIH (R01AR065174, U01AI119125). The UCSF-MS DNA biorepository is supported by RG-1611-26299 from the National Multiple Sclerosis Society to J.R.O. V.C. is supported by a Pfizer Chair Research Award, Rheumatology, University of Toronto. The Schroeder Arthritis Institute Psoriatic Disease Program is supported by the Krembil Foundation. M.A.S. acknowledges support from the Leo Foundation (LF-OC-22-001033). This research has been conducted using the UK Biobank Resource (approved project 15147). It uses data provided by patients and collected by the NHS as part of their care and support.

## Author contributions

J.T.E. and M.A.S. jointly supervised the study, which was conceived by J.T.E. and J.N.B. N.D. L.C.T., M.A.S. and J.T.E. contributed to study design. N.D., A.B., V.C., T.E., J.F., A.F., D.D.G., W.G., U.H., K.K., S.K., W.L., M.L., R. Mägi, R.P.N., P.R., A. Reis, C.H.S., J.N.B. and J.T.E. coordinated individual GWAS studies, with data preparation and association testing performed by N.D., P.E.S., J.B., D.E., J.N., M.T-L., L.F.T., T. Traks, S.U. and L.C.T. The meta-analysis was performed by N.D., with downstream bioinformatic

analysis by N.D., P.E.S., J.R.S. and L.C.T. J.E.G. and L.C.T. generated transcriptome data. P.D.M. and J.T.E. led the biological interpretation of results. G.A., D.B., F.B., A.C.B., M.A.B., H.B., F.C., R.C., C.J.C., M.D., E.E., O.F., S. Gerdes, C.E.M.G., S. Gulliver, P. Helliwell, P. Ho, P. Hoffmann, O.L.H., Z.H., K.H., D.J., M.K., C.K., C.L., S.H.L., F.M., S.K.M., N.M., R. McManus, E.H.M., M.J.N., M.N., V.O., J.R.O., M.T.P., B.E.P.W., A. Ramming, J.R., C.R., M.K.S., G.S., B.S., T. Tejasvi, H.T., J.J.V., E.M.W., R.B.W., R.W., S.W., X.W., Z.Z., the BSTOP study group and the Estonian Biobank research team contributed to the acquisition, generation and/or processing of data for individual GWAS or transcriptome studies. N.D., P.E.S., P.D.M., M.A.S. and J.T.E. drafted the manuscript and all authors critically reviewed the manuscript.

## Competing interests

F.C. reports grants and consultancy fees from Boehringer Ingelheim. S.K.M. reports departmental income from Abbvie, Almirall, Eli Lilly, Janssen, Leo Pharma, Novartis, Pfizer, Sanofi and UCB, outside the submitted work. M.J.N. has received consultancy fees and/or research funding from Abbvie, Amgen, Celgene, Eli Lilly, Janssen, Pfizer, Novartis and UCB. T. T. serves on an advisory board for L'Oreal Teledermatology. V.C. has received research grants from AbbVie, Amgen, and Eli Lilly and has received honoraria for advisory board member roles from AbbVie, Amgen, BMS, Eli Lilly, Janssen, Novartis, Pfizer, and UCB. His spouse is an employee of AstraZeneca. D.D.G. received grant support and/or consulting fees from Abbvie, Amgen, BMS, Eli Lilly, Janssen, Novartis, Pfizer and UCB. J.E.G. received research support from Eli Lilly, Kyowa Kirin, Janssen, Almirall, Celgene/BMS, Prometheus, Novartis, Galderma and AnaptysBio, and is a member of an advisory board for Novartis, AbbVie, Eli Lilly, Almirall, Galderma, Boehringer Ingelehim, Celgene/BMS, Sanofi, Janssen and AnaptysBio. S.K. is a founder of Prion OÜ, Geneto OÜ, Sportsgene OÜ and Genomic Therapeutics Pty Ltd. W.L. has received research grant funding from Abbvie, Amgen, Janssen, Leo, Novartis, Pfizer, Regeneron, and TRex Bio. P.D.M. reports consultancy fees from Unilever and speaker's fees from Sanofi and BMS. L.C.T. reports support from Janssen, Galderma, and Novartis. The remaining authors declare no competing interests.

## Additional information

Nick Dand [1,2], Philip E. Stuart [3], John Bowes [4,5], David Ellinghaus [6], Joanne Nititham [7], Jake R. Saklatvala [1], Maris Teder-Laving [8], Laurent F. Thomas [9,10,11,12], Tanel Traks [13], Steffen Uebe [14], Gunter Assmann [15,16], David Baudry [17], Frank Behrens [18,19,20,21], Allison C. Billi [3], Matthew A. Brown [1,22], Harald Burkhardt [19,20,21], Francesca Capon [1], Raymond Chung [23,24], Charles J. Curtis [23,24], Michael Duckworth [17], Eva Ellinghaus [6], Oliver FitzGerald [25], Sascha Gerdes [26], Christopher E. M. Griffiths [17,27,28], Susanne Gulliver [29], Philip S. Helliwell [30], Pauline Ho [4,5,31], Per Hoffmann [32], Oddgeir L. Holmen [33,34], Zhi-ming Huang [7], Kristian Hveem [10,33,34], Deepak Jadon [35,65], Michaela Köhm [18,19,20,21], Cornelia Kraus [14], Céline Lamacchia [36], Sang Hyuck Lee [23,24], Feiyang Ma [3,37], Satveer K. Mahil [17,38], Neil McHugh [39], Ross McManus [40], Ellen H. Modalsli [9,41], Michael J. Nissen [36], Markus Nöthen [32], Vinzenz Oji [42], Jorge R. Oksenberg [43], Matthew T. Patrick [3], Bethany E. Perez White [44], Andreas Ramming [45,46], Jürgen Rech [45,46], Cheryl Rosen [47], Mrinal K. Sarkar [3], Georg Schett [45,46], Börge Schmidt [48], Trilokraj Tejasvi [3,49], Heiko Traupe [42], John J. Voorhees [3], Eike Matthias Wacker [6], Richard B. Warren [50,51], Rachael Wasikowski [3], Stephan Weidinger [26], Xiaoquan Wen [52], Zhaolin Zhang [3], BSTOP study group*, Estonian Biobank research team*, Anne Barton [4,5,31], Vinod Chandran [53], Tõnu Esko [8], John Foerster [54], Andre Franke [6], Dafna D. Gladman [53], Johann E. Gudjonsson [3], Wayne Gulliver [29,55], Ulrike Hüffmeier [14], Külli Kingo [13,56], Sulev Kõks [57,58], Wilson Liao [7], Mari Løset [10,41], Reedik Mägi [59], Rajan P. Nair [3], Proton Rahman [60], André Reis [14], Catherine H. Smith [17,38], Paola Di Meglio [17], Jonathan N. Barker [17,38], Lam C. Tsoi [3,37,52], Michael A. Simpson [1,66] ✉ & James T. Elder [3,49,66] ✉

[1]Department of Medical & Molecular Genetics, School of Basic & Medical Biosciences, Faculty of Life Sciences & Medicine, King's College London, London, UK. [2]Health Data Research UK, London, UK. [3]Department of Dermatology, University of Michigan Medical School, Ann Arbor, MI, USA. [4]Centre for Genetics and Genomics Versus Arthritis, The University of Manchester, Manchester, UK. [5]National Institute for Health and Care Research (NIHR) Manchester Biomedical Research Centre, The University of Manchester, Manchester, UK. [6]Institute of Clinical Molecular Biology, Christian-Albrechts-University of Kiel, Kiel, Germany. [7]Deparment of Dermatology, University of California San Francisco, San Francisco, CA, USA. [8]Institute of Genomics, University of Tartu,

Tartu, Estonia. [9]Department of Clinical and Molecular Medicine, NTNU - Norwegian University of Science and Technology, Trondheim, Norway. [10]K.G. Jebsen Center for Genetic Epidemiology, Department of Public Health and Nursing, NTNU - Norwegian University of Science and Technology, Trondheim, Norway. [11]BioCore - Bioinformatics Core Facility, NTNU - Norwegian University of Science and Technology, Trondheim, Norway. [12]Clinic of Laboratory Medicine, St. Olavs Hospital, Trondheim University Hospital, Trondheim, Norway. [13]Department of Dermatology and Venereology, Institute of Clinical Medicine, University of Tartu, Tartu, Estonia. [14]Institute of Human Genetics, Universitätsklinikum Erlangen, FAU Erlangen-Nürnberg, Erlangen, Germany. [15]RUB University Hospital JWK Minden, Department of Rheumatology, Minden, Germany. [16]Jose-Carreras Centrum for Immuno- and Gene Therapy, University of Saarland Medical School, Homburg, Germany. [17]St John's Institute of Dermatology, School of Basic & Medical Biosciences, Faculty of Life Sciences & Medicine, King's College London, London, UK. [18]Division of Translational Rheumatology, Immunology - Inflammation Medicine, University Hospital, Goethe University, Frankfurt am Main, Germany. [19]Fraunhofer Institute for Translational Medicine and Pharmacology ITMP, Frankfurt am Main, Germany. [20]Fraunhofer Cluster of Excellence Immune-mediated Diseases CIMD, Frankfurt am Main, Germany. [21]Division of Rheumatology, University Hospital, Goethe University, Frankfurt am Main, Germany. [22]Genomics England, Canary Wharf, London, UK. [23]Institute of Psychiatry, Psychology and Neuroscience, King's College London, Denmark Hill, Camberwell, London, UK. [24]National Institute for Health and Care Research (NIHR) Biomedical Research Centre, South London and Maudsley Hospital, London, UK. [25]UCD School of Medicine and Medical Sciences and Conway Institute of Biomolecular and Biomedical Research, University College Dublin, Dublin, Ireland. [26]Department of Dermatology, Venereology and Allergy, University Hospital Schleswig-Holstein, Campus Kiel, Kiel, Germany. [27]Centre for Dermatology Research, University of Manchester, NIHR Manchester Biomedical Research Centre, Manchester, UK. [28]Department of Dermatology, King's College Hospital NHS Foundation Trust, London, UK. [29]Newlab Clinical Research Inc, St. John's, NL, Canada. [30]Leeds Institute of Rheumatic and Musculoskeletal Medicine, University of Leeds, Leeds, UK. [31]The Kellgren Centre for Rheumatology, Manchester University NHS Foundation Trust, Manchester, UK. [32]Institute of Human Genetics, University of Bonn, School of Medicine & University Hospital Bonn, Bonn, Germany. [33]HUNT Research Centre, Department of Public Health and Nursing, NTNU - Norwegian University of Science and Technology, Levanger, Norway. [34]Levanger Hospital, Nord-Trøndelag Hospital Trust, Levanger, Norway. [35]Department of Medicine, University of Cambridge, Cambridge, UK. [36]Division of Rheumatology, Geneva University Hospital, Geneva, Switzerland. [37]Department of Computational Medicine and Bioinformatics, University of Michigan, Ann Arbor, MI, USA. [38]St John's Institute of Dermatology, Guy's and St Thomas' National Health Service (NHS) Foundation Trust, London, UK. [39]Department of Life Sciences, University of Bath, Bath, UK. [40]Department of Clinical Medicine, Trinity Translational Medicine Institute, Trinity College Dublin, Dublin, Ireland. [41]Department of Dermatology, Clinic of Orthopedy, Rheumatology and Dermatology, St. Olavs Hospital, Trondheim University Hospital, Trondheim, Norway. [42]Department of Dermatology, University of Münster, Münster, Germany. [43]Weill Institute for Neurosciences, Department of Neurology, University of California, San Francisco, CA, USA. [44]Department of Dermatology, Northwestern University, Evanston, IL, USA. [45]Department of Internal Medicine 3, Friedrich-Alexander-University Erlangen-Nürnberg (FAU) and Universitätsklinikum Erlangen, Ulmenweg 18, 91054 Erlangen, Germany. [46]Deutsches Zentrum Immuntherapie (DZI), Friedrich-Alexander-University Erlangen-Nürnberg and Universitätsklinikum Erlangen, Erlangen, Germany. [47]Division of Dermatology, Toronto Western Hospital, University of Toronto, Toronto, Ontario, Canada. [48]Institute of Medical Informatics, Biometry and Epidemiology, University Hospital Essen, University of Duisburg-Essen, Essen, Germany. [49]Ann Arbor Veterans Affairs Hospital, Ann Arbor, MI, USA. [50]Division of Musculoskeletal and Dermatological Sciences, School of Biological Sciences, Faculty of Biology, Medicine and Health, The University of Manchester, Manchester, UK. [51]Centre for Dermatology Research, Salford Royal Hospital, Northern Care Alliance NHS Foundation Trust, Manchester Academic Health Science Centre, Manchester M6 8HD, UK. [52]Department of Biostatistics, Center for Statistical Genetics, University of Michigan, Ann Arbor, MI, USA. [53]Schroeder Arthritis Institute, Krembil Research Institute and Toronto Western Hospital, University Health Network and University of Toronto, Toronto, Ontario, Canada. [54]College of Medicine, Dentistry, and Nursing, University of Dundee, Dundee, UK. [55]Department of Dermatology, Discipline of Medicine, Faculty of Medicine, Memorial University of Newfoundland, St. John's, NL, Canada. [56]Dermatology Clinic, Tartu University Hospital, Tartu, Estonia. [57]Perron Institute for Neurological and Translational Science, Nedlands, WA 6009, Australia. [58]Centre for Molecular Medicine and Innovative Therapeutics, Health Futures Institute, Murdoch University, Perth, WA 6150, Australia. [59]Estonian Genome Centre, Institute of Genomics, University of Tartu, Tartu, Estonia. [60]Memorial University of Newfoundland, St. John's, NL, Canada. [65]Deceased: Deepak Jadon. [66]These authors jointly supervised this work: Michael A. Simpson, James T. Elder. *Lists of authors and their affiliations appear at the end of the paper. ✉e-mail: michael.simpson@kcl.ac.uk; jelder@umich.edu

## BSTOP study group

A. David Burden[61], Catherine H. Smith[17], Jonathan N. Barker[17,38], Sara J. Brown[62,63], Nick Dand[1,2], Satveer K. Mahil[38], Helen McAteer[64], Julia Schofield[64] & Stefan Siebert[61]

[61]School of Infection and Immunity, University of Glasgow, Glasgow, UK. [62]Centre for Genomic and Experimental Medicine, Institute of Genetics and Cancer, University of Edinburgh, Edinburgh, UK. [63]Department of Dermatology, NHS Lothian, Edinburgh, UK. [64]Psoriasis Association, Northampton, UK.

## Estonian Biobank research team

Tõnu Esko[8], Andres Metspalu[59], Lili Milani[59], Reedik Mägi ⑩ [59] & Mari Nelis[59]

