## [Transparent Peer Review file · Nature Communications]

GWAS meta-analysis of psoriasis identifies new susceptibility alleles impacting disease mechanisms and therapeutic targets

Corresponding Author: Dr James Elder

Version 0:

Reviewer comments:

Reviewer #1

(Remarks to the Author)

Summary:

Dand et al. conducted a meta-analysis of psoriasis GWAS across 18 cohorts, including 36,466 cases. The authors found 45 novel associations, followed by a standard set of GWAS downstream analyses, including fine mapping, genetic correlation, TWAS, and joint analysis with a single-cell dataset of psoriasis-affected skin. Generally, their findings are solid and have a lot of scientific value, but I feel there are multiple points to be improved. Please see my comments below.

Major comments:

- 1, Supplemental Table 1 nicely summarizes their GWAS results and would be a valuable resource for future studies. I recommend discussing and adding heterogeneity in effect size estimates across cohorts. Since their meta-analysis model assumes that the effect sizes are “fixed” across cohorts, the heterogeneity stats (e.g., Cochran Q value) should not be large. Even if they only used cohorts of European ancestries, there might be clinical heterogeneity across cohorts. If they observe substantial heterogeneities, they need to discuss the validity of their analytical strategy.
- 2, Line 176-180. They discussed the replicability of previously reported psoriasis-associated variants. In addition to the existence of association, they should report effect size consistency. Are their associations show consistency in terms of signs and magnitudes?
- 3, Related to comment 2, across-ancestral effect size comparison is also important. In line 185, they stated, “Five of these 50 loci encompass variants previously reported at genome-wide significance in other ancestral populations”. They should discuss this point.
- 4, Regarding TWAS, they applied a standard pipeline using the S-PrediXan pipeline. However, several articles reported critical limitations of TWAS, so they need to be very cautious about the interpretation (such as 30926968). For example, even when the true causal biology of a locus is not eQTL (such as a missense variant), TWAS might falsely provide a strong association. Moreover, since the causal genes tend to be close to the causal variant (not necessarily the closest gene; PMID: 33828297) and TWAS usually prioritizes close genes, TWAS might nominate the true causal genes simply by chance. Therefore, looking at the gene names does not provide robust evidence supporting its validity. I have two suggestions.
 - 4-1. Please apply mediated expression score regression (MESC) (PMID: 32424349). By doing so, the authors can estimate how much of the eQTLs mediate GWAS causal signals.
 - 4-2. They need to pay attention to the signs of the TWAS effects. Sup-Table 10 shows the downregulation of IL23A, the downregulation of STAT3, and the downregulation of TYK2 are related to the increased genetic risk of psoriasis (they have “negative” TWAS effects). I think these signs are the opposite to the author’s expectations.
- 5, They nicely connected single-cell transcriptome data with GWAS results. There are several peer-reviewed methods for this aim (such as sc-linker; PMID: 36175791). They also need to apply them since the readers are unsure about the validity

of the author's analysis strategy.

6. Related to comments 4 and 5, could they compare the genetically predicted gene expression modification (ie. TWAS effect "signs") and biological gene expression modifications (ie. differentially expressed gene "signs" in the single cell gene expression or transcriptome data following a series of cytokine challenges discussed in Supplementary Figure 9)? As reported in Marigorta et al (PMID 28805827), are there many coherent genes? Or many incoherent genes? If the authors find an excess of coherent genes, they can claim the validity of TWAS results.

Minor comments:

1, In the abstract, they stated "saturation map". It might be an over-statement since the detected variants in this study only explain 14.6% of common variant heritability (Sup-Table 4), this is far from saturation.

2, Line 176: "Genome-wide significant psoriasis associations were also observed in the present study at all but two of the previously reported susceptibility loci in Europeans (Figure 1, Supplementary Table 1)." They discussed the replicability of previously reported associations. It is not very clear what exactly they analyzed here. Please improve the clarity of this part.

Reviewer #2

(Remarks to the Author)

This is a well-conducted and well written manuscript that has several important discoveries and will be of broad interest to the scientific community.

The authors have skillfully used cutting edge methods to obtain and interpret GWAS results. They have presented their findings with clear and logical writing.

I only have a few minor suggestions aimed at highlighting some of the more salient lessons.

First, their analysis has identified evidence implicating regulatory and coding variants that identify specific genes. This discovery provides opportunities to experimentally improve our understanding of disease mechanisms and provides a starting point for prioritizing clinically relevant genes and variants to advance clinical sequencing in psoriasis management. It also creates an opportunity to resolve etiological heterogeneity, which would also help to improve disease management. Therefore, I would suggest adding a small table to the main manuscript summarizing the genes identified by each method. For example, "Chr-LD block -With protein-altering variants-TWAS candidates" for the 15 loci with one or more genes implicated. If there are space constraints, you could just focus on the subset of 10 loci where the evidence converges from both methods.

Also, to be instructive and explicit about why larger GWAS are important, I suggest that the authors add a sentence to the paragraph starting on line 219 page 7 about the distributions of allele frequencies and odds ratios for the protein coding genes. I would guess that they are on the lower end of AF for GWAS variants, and thus highlight that large cohorts are needed and a continued investment in further GWAS will aid in the clinical translation of discoveries. I might expect ORs could be towards the higher end of GWAS variants, which suggests that they will improve prediction of disease risk.

The discussion related to the therapeutic implications of their findings could include a discussion about the implications from their causal inference studies. For example, a causal role of adiposity in psoriasis suggests that therapeutic interventions aimed at reducing adiposity should help to ameliorate psoriasis symptoms for some individuals with psoriasis.

The discussion should mention the lack of ancestral diversity as a limitation why it is important to improve diversity in future psoriasis GWAS.

Reviewer #3

(Remarks to the Author)

Dand et al. presented a large-scale GWAS meta-analysis of 18 studies comprising 36,466 cases and 458,078 controls for psoriasis in this submission. They identified 109 psoriasis risk loci including 45 previously unreported ones and characterized putative causal variants, genes, pathways, and cellular context, which together improve the biology of psoriasis. The study was rigorously performed, and the manuscript was concisely written. The reviewer raised several concerns as below:

1. To quantify the scale that the observed genomic inflation factor was explained by the polygenicity, the reviewer would recommend providing the ratio of LDSC intercept and mean chi-squared which is reported in LDSC in line 167.
2. The supplementary tables were NOT ordered by their appearance in the manuscript. For example, the reviewer would recommend switching the current supplementary table 17 to supplementary table 1.
3. The reviewer did not find the procedures and criteria applied for genotyping, genotype imputation, QC, and GWAS in each of the 18 contributing studies in the current supplementary table 17.
4. The authors argued that they identified duplicates and close relatives between the contributing studies using a relatively small set of genotyping markers. It would be useful to cite responding reference(s) to support that the number of markers is

sufficient for estimating kinships.

5. Since the authors used in the meta-analysis the one with highest imputation quality for the same SNVs across multiple imputation panels, how would that procedure influence the downstream COJO or the statistical fine-mapping analyses? Have the authors evaluated the influence?

6. It was nice to compare the fine-mapping resolution between the 2017 and the current studies. In the comparison, have the authors used the same set of variants as input?

7. The conclusion "We found strong evidence that the well-known psoriasis-associated missense variant rs33980500 (p.Asp10Asn; Figure 2B) itself has a regulatory role, and independently that rs6908585 is a skin-specific regulatory variant (regulatory probability of 0.491; Figure 2C), suggesting that both coding and gene regulation of TRAF3IP2 contribute to psoriasis susceptibility at this locus" in lines 245-249 does not sound solid to the reviewer. Does the authors have promising evidence to support rs33980500 affects psoriasis risk through altering protein rather than a regulatory effect?

8. Of the genes identified in TWAS, how many achieved significant colocalization?

9. For the 12 traits having potential causal relationship with psoriasis (line 327-329), have the authors evaluated the causal relationship in Mendelian randomization? If so, how many of them achieve consistent results in Mendelian randomization?

Version 1:

Reviewer comments:

Reviewer #1

(Remarks to the Author)

Summary:

I think the authors fairly addressed the points I raised, and I am satisfied with the authors' responses. I have two minor points.

Minor comments:

1. Regarding "Effect size estimates were consistent with the previous psoriasis meta-analysis⁷ (Supplementary Figure 2)", the replicability is beautiful. I want to confirm whether there are any shared samples between the current study (Y-axis) and the 2017 meta-analysis (X-axis). If there are too many overlapped samples, this is not informative at all.

2. I have a comment regarding their response: "We thank the reviewer for this comment. We would like to clarify that the aim of our analysis was different to that of sc-linker, where the aim is to identify causal genes. Instead, we have used scRNA-seq data to observe the cellular expression distributions for the genes identified by our TWAS analysis. This process is analogous to cell marker identification, allowing us to describe expression differences across skin cell types for these TWAS genes (for which the reference panels were based on bulk RNA-seq data) and consequently to identify cell types with relatively abundant expression of TWAS target genes in psoriatic skin. We have now clarified this objective of the analysis in our Results section". I think the authors misunderstood the value of sc-linker. Sc-linker is an analytical pipeline integrating single-cell RNA-sequencing, epigenomic SNP-to-gene maps, and GWAS summary statistics to infer the underlying cell types by which genetic variants influence disease. So, sc-linker aims to identify causal cell types, not causal genes. The authors sought to identify cell types that express large amounts of TWAS-nominated genes, so their aim is similar to sc-linker. However, the difference is that sc-linker is a typical polygenic analysis (cumulative effect of genome-wide weak associations) using S-LDSC, whereas the authors' approach relied on a few loci with strong associations. Therefore, I believe the sc-linker approach is complementary to their approach and potentially enriches their depth of investigation. This is why I recommend clinker analysis.

Reviewer #3

(Remarks to the Author)

The authors have addressed all of the reviewer's concerns in the revised manuscript.

REVIEWER COMMENTS

We thank the editor and reviewers for reviewing our manuscript and for the helpful suggestions. We have made a number of changes to the manuscript to directly address many of these comments, as detailed in a point-by-point response below.

Reviewer #1 (Remarks to the Author):

Summary:

Dand et al. conducted a meta-analysis of psoriasis GWAS across 18 cohorts, including 36,466 cases. The authors found 45 novel associations, followed by a standard set of GWAS downstream analyses, including fine mapping, genetic correlation, TWAS, and joint analysis with a single-cell dataset of psoriasis-affected skin. Generally, their findings are solid and have a lot of scientific value, but I feel there are multiple points to be improved. Please see my comments below.

Thank you for taking the time to review our manuscript. We appreciate the positive comments.

Major comments:

1, Supplemental Table 1 nicely summarizes their GWAS results and would be a valuable resource for future studies. I recommend discussing and adding heterogeneity in effect size estimates across cohorts. Since their meta-analysis model assumes that the effect sizes are “fixed” across cohorts, the heterogeneity stats (e.g., Cochran Q value) should not be large. Even if they only used cohorts of European ancestries, there might be clinical heterogeneity across cohorts. If they observe substantial heterogeneities, they need to discuss the validity of their analytical strategy.

We agree that heterogeneity in effect sizes is expected due to the different ascertainment strategies employed by contributing studies, and indeed have demonstrated this in previous work (PMID: 36870556). We have added heterogeneity statistics to Supplementary Table 1 (we include I^2 instead of the Q statistic because I^2 should be comparable across variants with effect size estimates from differing numbers of underlying studies; we also include the Cochran Q test p-value). We have added the following statement to the Results, along with new Supplementary Figures:

“Effect size heterogeneity was observed at six of the 109 lead variants after accounting for multiple tests ($P_{\text{het}} < 4.6 \times 10^{-4}$), corresponding to relatively large effect loci (risk allele OR > 1.15) (Supplementary Figure 3). A modest degree of heterogeneity is expected due to the range of ascertainment approaches across studies²⁶, with population-based biobank GWAS consistently exhibiting attenuated effect sizes (Supplementary Figure 4).”

We also add the following to the Discussion:

“We note that differing ascertainment strategies among the constituent studies, with consequent variation in clinical presentation and severity of sampled psoriasis cases, likely contribute to the effect size heterogeneity observed at six loci and slightly attenuated effect size estimates at previously established European loci (Supplementary Figure 2).”

We intend to include heterogeneity statistics in the full summary statistics that will be deposited in GWAS Catalog at the point of publication.

2, Line 176-180. They discussed the replicability of previously reported psoriasis-associated variants. In addition to the existence of association, they should report effect size consistency. Are their associations show consistency in terms of signs and magnitudes?

The effect sizes are consistent with those previously reported, albeit with a slight attenuation that is likely due to a combination of winner's curse and the inclusion of general population ascertained cases with lower average severity than specialist ascertained cases. The attenuation of effect sizes is covered in the edits made in response to the reviewer's previous comment. In addition, we have added a new Supplementary Figure, and have added the following to the Results section:

"Effect size estimates were consistent with the previous psoriasis meta-analysis⁷ (Supplementary Figure 2)."

3, Related to comment 2, across-ancestral effect size comparison is also important. In line 185, they stated, "Five of these 50 loci encompass variants previously reported at genome-wide significance in other ancestral populations". They should discuss this point.

We agree that this point is worth expanding. On closer inspection we identified an error in the position of one of the previously reported Han Chinese associations, which now maps outside of our susceptibility loci and leaves four of the 50 new European loci with previous reports in other populations. The results and Supplementary Table 1 have been adjusted accordingly. We have also expanded on the source populations and estimated effect size of the previously reported associations, and have additionally considered non-European susceptibility variants that do not fall within any of the 109 loci that we report in the current manuscript. We add the following text to the Results:

"Four of these 50 loci encompass variants previously reported at genome-wide significance in other ancestral populations, either in Han Chinese studies (chromosomes 1p36.22, 2q11.2 and 4q27, albeit the reported lead variants are not associated in the current study)^{24, 25} or a trans-ancestry meta-analysis including European and South Asian samples (1p36.22 and 1q24.2, with consistent effects in the current study)²² (Supplementary Table 4)."

And in the Discussion:

"Furthermore, 24 susceptibility loci reported in non-European populations were not observed with genome-wide significant evidence of association (Supplementary Table 21, Supplementary Figure 17). Inter-population differences in allele frequency or local LD structure may partially account for this observation, with 11 (46%) of the lead variants less than 500 kb from a variant with suggestive evidence of association ($P < 1.0 \times 10^{-5}$) in the current study. The remaining 13 (54%) may represent ancestry-specific associations, and further studies are needed to address the important question of how genetic variation influences psoriasis risk across ancestry groups.^{75, 76}"

4, Regarding TWAS, they applied a standard pipeline using the S-PrediXan pipeline. However, several articles reported critical limitations of TWAS, so they need to be very cautious about the interpretation (such as 30926968). For example, even when the true causal biology of a locus is not eQTL (such as a missense variant), TWAS might falsely provide a strong association. Moreover, since the causal genes tend to be close to the causal variant (not necessarily the closest gene; PMID: 33828297) and TWAS usually prioritizes close genes, TWAS might nominate the true causal genes simply by chance. Therefore,

looking at the gene names does not provide robust evidence supporting its validity. I have two suggestions.

4-1. Please apply mediated expression score regression (MESC) (PMID: 32424349). By doing so, the authors can estimate how much of the eQTLs mediate GWAS causal signals.

4-2. They need to pay attention to the signs of the TWAS effects. Sup-Table 10 shows the downregulation of *IL23A*, the downregulation of *STAT3*, and the downregulation of *TYK2* are related to the increased genetic risk of psoriasis (they have "negative" TWAS effects). I think these signs are the opposite to the author's expectations.

Thank you for raising these points, and we fully agree that TWAS findings need to be interpreted carefully.

Regarding comment 4-1, we have undertaken MESC analysis and reassuringly obtained results consistent with other complex disease traits. These are described in the new Supplementary Table 11, with a corresponding addition to the Methods section and the following statement added to Results:

“we estimated the degree to which the contribution to psoriasis risk of common genetic variation is mediated through effects on gene expression. We found 32.4% of heritability to be mediated by the *cis* genetic component of gene expression levels across tissues, with between 13.7% and 18.3% in blood and skin (Supplementary Table 11), estimates that are broadly consistent with other complex disease traits.³⁴”

Regarding comment 4-2, as described in further detail in response to comment 8 of Reviewer #3, we have performed additional filtering of TWAS genes to leave a shorter list of genes that are more robustly predicted to be differentially regulated in the presence of psoriasis risk variants. *IL23A* and *TYK2* do not meet the more stringent criteria and are now excluded from the TWAS gene list. However, the more general point to pay attention to TWAS effect directions remains valid.

The TWAS prediction models use information from multiple genetic variants around and within each gene to make predictions. By comparing against a simple lookup of lead psoriasis variants in GTEx eQTL data, we have confirmed that the more complex TWAS prediction models are, in general, not introducing highly unexpected effect directions:

For *STAT3* specifically, the estimated TWAS effect size is -0.2 in sun-exposed skin. While the overall lead psoriasis susceptibility variant at this locus is not present in the GTEx eQTL data, the lead fine-mapped variant at the strongest of the independent signals is present, and the psoriasis risk allele has a negative association with expression (slope = -0.09), consistent with the TWAS effect direction.

We have also considered whether the TWAS effect directions align with our expectations from previously published RNA-seq differential expression (DE) results in bulk skin (PMID: 30641038). Since the TWAS prediction models are based on healthy GTEx participant samples, and because psoriasis lesions exhibit an acute inflammatory transcriptomic signature, we considered the most appropriate comparison to be against differential expression between psoriasis non-lesional and healthy control skin. We observe that one TWAS gene (*LCE3A*) is significantly upregulated in non-lesional skin, in

agreement with the effect direction predicted by TWAS, where we note it has the largest and second-largest effect estimates among all significant genes for non-exposed and exposed skin, respectively. No other TWAS genes are observed to be differentially expressed (and it follows that none, including *STAT3*, are observed to be differentially expressed in the opposite direction to the TWAS prediction). There are several reasons why we may not see more fully overlapping effects between TWAS predictions and the published DE data, including: (i) the relatively small sample size of the published DE data, which is based on 28 non-lesional psoriasis samples and 38 healthy control samples, limiting power to detect differentially expressed genes; (ii) the fact that even non-lesional skin in psoriasis patients does manifest the transcriptomic effects of systemic inflammation (PMID: 30641038), and thus differences relative to healthy skin are not precisely analogous to the effects predicted by TWAS, which might be interpreted as transcriptomic differences between the skin of genetically predisposed and non-predisposed individuals prior to disease onset; (iii) the fact that observed differences in expression will result from environmental factors (at both the cellular and organism level) in addition to genetic regulation. Overall, we conclude that the TWAS results provide important candidate genes for further investigation, provided that findings are interpreted with sufficient caution. As such, we have expanded our Discussion to highlight this:

“Finally, our TWAS analysis relies on reference data from healthy participants to link susceptibility-associated genetic variation to predicted expression differences in skin and blood. TWAS have demonstrated utility in prioritizing causal genes but nonetheless harbour the potential for false positive findings when applied to incorrect tissues.⁷⁷ While we provide additional context for our TWAS candidate genes in psoriatic single-cell expression data and in cytokine-stimulated keratinocytes, future studies are needed to map causal relationships from psoriasis risk alleles to variation in the transcriptomic profiles of psoriasis patients undergoing systemic inflammation.”

5, They nicely connected single-cell transcriptome data with GWAS results. There are several peer-reviewed methods for this aim (such as sc-linker; PMID: 36175791). They also need to apply them since the readers are unsure about the validity of the author’s analysis strategy.

We thank the reviewer for this comment. We would like to clarify that the aim of our analysis was different to that of sc-linker, where the aim is to identify causal genes. Instead, we have used scRNA-seq data to observe the cellular expression distributions for the genes identified by our TWAS analysis. This process is analogous to cell marker identification, allowing us to describe expression differences across skin cell types for these TWAS genes (for which the reference panels were based on bulk RNA-seq data) and consequently to identify cell types with relatively abundant expression of TWAS target genes in psoriatic skin. We have now clarified this objective of the analysis in our Results section:

“To identify the cellular contexts of genes whose transcription is mediated by psoriasis susceptibility variation, we asked which cells express TWAS target genes abundantly in the skin of psoriasis patients.”

6. Related to comments 4 and 5, could they compare the genetically predicted gene expression modification (ie. TWAS effect "signs") and biological gene expression modifications (ie. differentially expressed gene "signs" in the single cell gene expression or transcriptome data following a series of cytokine challenges discussed in Supplementary Figure 9)? As reported in Marigorta et al (PMID 28805827), are there many coherent genes? Or many incoherent genes? If the authors find an excess of coherent genes, they can claim the validity of TWAS results.

We have considered this suggestion carefully. We feel that the most directly relevant comparison to make is against bulk-tissue differential expression observed between non-lesional psoriasis skin and healthy controls. As described above in the response to comment 4, we see consistent/coherent expression differences for *LCE3A* in skin, and no evidence of incoherent expression differences. We describe above several reasons that there may not be more consistency with the published expression differences and the related text that we have added to the Discussion.

Minor comments:

1, In the abstract, they stated “saturation map”. It might be an over-statement since the detected variants in this study only explain 14.6% of common variant heritability (Sup-Table 4), this is far from saturation.

We agree with the sentiment here and have replaced “to move towards a saturation map of psoriasis susceptibility...” with “to refine the genetic map of psoriasis susceptibility...”.

2, Line 176: “Genome-wide significant psoriasis associations were also observed in the present study at all but two of the previously reported susceptibility loci in Europeans (Figure 1, Supplementary Table 1).” They discussed the replicability of previously reported associations. It is not very clear what exactly they analyzed here. Please improve the clarity of this part.

Thank you for pointing this out and we agree that more clarity is needed. Since this is quite a technical point, we now refer the reader to the Methods where we have elaborated as follows:

“We considered previously reported associations to be recapitulated if they fell within the same LD block as a genome-wide significant signal (i.e. lead variant) in the current study. Note that association signals previously reported as distinct loci but falling within the same LD block are classed as a single locus in the current study. This occurred at four (current study) loci, albeit in each case multiple independent signals were identified (Supplementary Table 1). Effect size consistency against the previous (2017) meta-analysis⁷ was assessed based on the lead variants at previous meta-analysis loci, using marginal effect estimates and without regard to their correspondence to primary or secondary association signals within newly defined loci.”

To further improve the clarity of this part of the Methods we have changed the section heading from “Associated regions and independent signal identification” to “Definition of susceptibility loci and independent signal identification” and made minor revisions to the language that are marked in the resubmitted manuscript.

Finally, the column headings for Supplementary Table 2 (Previously reported European psoriasis susceptibility loci not found genome-wide significantly associated) have been updated to make it completely clear that the values in the table represent effect estimates from the current study.

Reviewer #2 (Remarks to the Author):

This is a well-conducted and well written manuscript that has several important discoveries and will be of broad interest to the scientific community.

The authors have skillfully used cutting edge methods to obtain and interpret GWAS results. They have presented their findings with clear and logical writing.

We very much appreciate the positive remarks; thank you for your time in considering our manuscript.

I only have a few minor suggestions aimed at highlighting some of the more salient lessons.

First, their analysis has identified evidence implicating regulatory and coding variants that identify specific genes. This discovery provides opportunities to experimentally improve our understanding of disease mechanisms and provides a starting point for prioritizing clinically relevant genes and variants to advance clinical sequencing in psoriasis management. It also creates an opportunity to resolve etiological heterogeneity, which would also help to improve disease management. Therefore, I would suggest adding a small table to the main manuscript summarizing the genes identified by each method. For example, “Chr-LD block -With protein-altering variants-TWAS candidates” for the 15 loci with one or more genes implicated. If there are space constraints, you could just focus on the subset of 10 loci where the evidence converges from both methods.

We agree with the reviewer that a main summary table would be valuable. We have now added Table 1, which describes the lead variant and candidate genes for the 15 newly reported loci at which there is evidence of protein-altering variants, expression differences predicted via TWAS, or an existing psoriasis drug target.

Also, to be instructive and explicit about why larger GWAS are important, I suggest that the authors add a sentence to the paragraph starting on line 219 page 7 about the distributions of allele frequencies and odds ratios for the protein coding genes. I would guess that they are on the lower end of AF for GWAS variants, and thus highlight that large cohorts are needed and a continued investment in further GWAS will aid in the clinical translation of discoveries. I might expect ORs could be towards the higher end of GWAS variants, which suggests that they will improve prediction of disease risk.

Thank you for this suggestion. We have investigated the reviewer’s hypotheses and our data broadly support them, especially when considering the subset of coding variants with strongest deleteriousness predictions. Minor allele frequency shows a tendency to be lower among 47 coding variants than 5,295 other variants in 95% credible sets, albeit not statistically significant ($p=0.41$, Mann-Whitney test; $p=0.051$ when further subdividing coding variants by deleteriousness [CADD score >15 vs. ≤ 15], Kruskal-Wallis rank sum test). Effect size is significantly higher among coding variants ($p=0.004$, Mann-Whitney; $p=0.0002$ when further subdividing, Kruskal Wallace). We agree with the reviewer’s suggestion that increasingly large GWAS studies are needed and justified; however, since increasing effect size and reducing allele frequency will affect statistical power for detection in opposite directions, we feel these results don’t clearly illustrate the suggestion. Instead, we have compared the allele frequencies and effect sizes observed at the new loci to those at previously known loci, illustrating this with a new Supplementary Figure 6 and adding the following to the results paragraph in question:

“The fact that these candidate coding variants are identified in novel loci, at which moderate-to-large effect susceptibility variants are detected at lower minor allele frequencies than for known loci (Supplementary Figure 6), provides further justification that expanded psoriasis meta-analyses will continue to generate translational insights.”

The discussion related to the therapeutic implications of their findings could include a discussion about

the implications from their causal inference studies. For example, a causal role of adiposity in psoriasis suggests that therapeutic interventions aimed at reducing adiposity should help to ameliorate psoriasis symptoms for some individuals with psoriasis.

We agree, and indeed some of us have recently reviewed the relevant observational literature. We have now rectified this omission with the following discussion:

“Our results also have implications for non-pharmacological interventions to manage psoriasis. We present evidence that increased adiposity, measured by body mass index, fat mass or waist circumference, leads to elevated psoriasis risk. This finding is consistent with previous observational studies demonstrating that psoriasis incidence and severity in obese individuals can be reduced through adiposity-reducing lifestyle interventions⁵⁴, and with observed improvements in comorbid psoriasis in type 2 diabetes patients treated with weight loss-inducing glucagon-like peptide 1 (GLP-1) receptor agonists.⁵⁵ Both adiposity and metabolic profile, the latter congruent with the causal role of triglycerides that our study highlights, may mediate the hypothesized effects of diet on psoriasis severity that are the subject of several ongoing trials.^{56, 57}”

The discussion should mention the lack of ancestral diversity as a limitation why it is important to improve diversity in future psoriasis GWAS.

We fully agree with the reviewer, and as highlighted in response to Reviewer #1’s related comment, we have expanded the Discussion:

“Furthermore, 24 susceptibility loci reported in non-European populations were not observed with genome-wide significant evidence of association (Supplementary Table 21, Supplementary Figure 17). Inter-population differences in allele frequency or local LD structure may partially account for this observation, with 11 (46%) of the lead variants less than 500 kb from a variant with suggestive evidence of association ($P < 1.0 \times 10^{-5}$) in the current study. The remaining 13 (54%) may represent ancestry-specific associations, and further studies are needed to address the important question of how genetic variation influences psoriasis risk across ancestry groups.^{75, 76}”

Reviewer #3 (Remarks to the Author):

Dand et al. presented a large-scale GWAS meta-analysis of 18 studies comprising 36,466 cases and 458,078 controls for psoriasis in this submission. They identified 109 psoriasis risk loci including 45 previously unreported ones and characterized putative causal variants, genes, pathways, and cellular context, which together improve the biology of psoriasis. The study was rigorously performed, and the manuscript was concisely written. The reviewer raised several concerns as below:

We are grateful for the detailed review of our submission, and appreciate the positive comments and further suggestions.

1. To quantify the scale that the observed genomic inflation factor was explained by the polygenicity, the reviewer would recommend providing the ratio of LDSC intercept and mean chi-squared which is reported in LDSC in line 167.

We have updated the indicated sentence to incorporate our conclusions based on the LDSC ratio:

“The genomic inflation factor²⁰ (λ_{GC}) of 1.14 and LD score regression intercept of 1.07 indicate modest inflation of the meta-analysis test statistics that is primarily driven by polygenicity (estimated proportion ascribed to other causes: 0.15), consistent with other complex diseases.^{21”}

We also clarify in the Methods that this value is the LDSC ratio:

“LD score regression intercept and the proportion of inflation attributed to causes other than polygenic heritability (“LDSC ratio”) were calculated using LDSC software with default settings and precomputed LD scores derived from 1000 Genomes data.^{21”}

2. The supplementary tables were NOT ordered by their appearance in the manuscript. For example, the reviewer would recommend switching the current supplementary table 17 to supplementary table 1.

We agree that the originally submitted Supplementary Table 17 (now renumbered Supplementary Table 22) contains details relating to methodology, which might conceptually be read before other supplementary tables. However, since the journal style has Methods after Results and Discussion, we believe this is correct. As such, we have not reordered supplementary tables, but would of course be happy to do so if the editors feel we have misinterpreted the guidance.

3. The reviewer did not find the procedures and criteria applied for genotyping, genotype imputation, QC, and GWAS in each of the 18 contributing studies in the current supplementary table 17.

Apologies for any confusion on this point. A detailed description of methods employed for each individual GWAS study was provided in a separate Supplementary Methods Table. To simplify the presentation, we have now moved this to Supplementary Table 23.

4. The authors argued that they identified duplicates and close relatives between the contributing studies using a relatively small set of genotyping markers. It would be useful to cite responding reference(s) to support that the number of markers is sufficient for estimating kinships.

Thank you for raising this point. To establish that the number of markers should be sufficient for robust identification of duplicates and close relatives, we performed testing on the BSTOP dataset for which full genotype data were available in-house. We considered the duplicates, first- and second-degree relationships identified with a maximal set of 73,223 LD-independent markers outside previously reported psoriasis loci and regions of long-range LD to represent the ground truth (i.e., best possible with full genotype data available). We then randomly selected smaller subsets of markers to assess our ability to accurately recover these relationships. We found that under KING’s default relationship inference settings, all 10 relationships were recovered down to approx. 2500 markers, with at most two “false positive” relationships. We therefore specified a target overlap of 5000 markers between cohorts, albeit due to limited overlap between the arrays used in some studies it was necessary to drop below this target in some cases. We have now added a section to the Supplementary Methods describing this process.

5. Since the authors used in the meta-analysis the one with highest imputation quality for the same SNVs across multiple imputation panels, how would that procedure influence the downstream COJO or the statistical fine-mapping analyses? Have the authors evaluated the influence?

We designed the LD reference panel used by the downstream COJO fine-mapping analyses in a way that attempted to minimize any bias or adverse influence stemming from the use of multiple imputation panels for the meta-GWAS. As recommended by the authors of GCTA-COJO (PMID: 22426310) and discussed in the Supplementary Methods of the manuscript, the LD reference panel was based on a large representative sample of >24,000 European ancestry individuals that were a substantial subset of the 18 European-ancestry studies in the meta-GWAS. Furthermore, genotypes of panel variants were imputed and selected in a manner identical to that followed by 15 of the 18 studies of the meta-GWAS, i.e., imputing variants twice with the 1000 Genomes (phase 3) and HRC reference panels and selecting on a per variant basis genotypes from the reference panel with the higher imputation quality. As detailed in Supplementary Table 23, two of the three meta-GWAS studies that did not follow this precise procedure also employed in part the 1000 Genomes (phase 3) and/or HRC panels.

COJO does not provide any diagnostics to verify if a constructed LD reference panel is sufficiently large and unbiased to adequately characterize the LD structure of the population sampled for the meta-GWAS. However, another genetic fine-mapping method, susieR (PMID: 37220626, 35853082), uses the same inputs of association summary statistics and LD correlation matrix as COJO, but offers several diagnostics including the function “estimate_s_rss”, which estimates a parameter s measuring the consistency between the observed association Z-scores and the LD matrix. Parameter s is a real number between 0 and 1; small s values indicate good consistency between the observed Z-scores and the LD matrix. We estimated parameter s using version 0.12.27 of susieR for each of the 104 LD blocks from our 90% N_{eff} -filtered meta-GWAS that had at least one genome-wide significant association signal. The histogram below summarizes the distribution of the resulting estimated s values:

The mean and median for this distribution are 0.0022 and 0.0018, respectively, and the range is 0.00026–0.013. These uniformly small s values indicate very good consistency between the association summary statistics of our meta-GWAS and the LD reference panel that we used for downstream statistical fine-mapping with COJO.

6. It was nice to compare the fine-mapping resolution between the 2017 and the current studies. In the comparison, have the authors used the same set of variants as input?

We did not use the same set of variants as input for the fine-mapping comparison. As mentioned in the “Statistical fine-mapping” subsection of the Methods section, for each study we used all variants within 200kb of the 2017 lead markers of 63 susceptibility loci that were well-imputed for a $N_{\text{eff}} > 90\%$ in their respective meta-analysis ($N_{\text{eff}} > 27,537$ for the 2017 study and $N_{\text{eff}} > 93,252$ for the current study). The overlap in the two sets of variants used for the fine-mapping comparison is shown by this Venn diagram:

No. variants used for 95% BCS analysis of 63 unconditional PsV loci

We have rerun the fine-mapping comparison, restricting the analysis for the two meta-studies to the shared set of 52,871 variants. Using the shared set of variants had only a modest effect on the resulting 95% Bayesian credible sets:

parameter	2017 meta-analysis		Current meta-analysis	
	all variants	shared variants	all variants	shared variants
No. analyzed variants	70,705	52,871	64,091	52,871
Median no. 95% BCS variants	16	14	8	8
Median length of 95% BCS (bp)	46,773	43,532	25,495	25,495
No. 95% BCS with ≤ 5 variants	12	12	23	25
No. 95% BCS with max PP ≥ 0.50	10	9	19	17

The conclusion we made in the Results section concerning the fine-mapping comparison remains unchanged when based on an analysis restricted to shared variants; namely, compared to the 2017 meta-analysis, the current meta-analysis revealed the same or fewer number of variants in 95% credible sets at 52 (83%) of the 63 established psoriasis susceptibility loci.

7. The conclusion "We found strong evidence that the well-known psoriasis-associated missense variant rs33980500 (p.Asp10Asn; Figure 2B) itself has a regulatory role, and independently that rs6908585 is a skin-specific regulatory variant (regulatory probability of 0.491; Figure 2C), suggesting that both coding and gene regulation of TRAF3IP2 contribute to psoriasis susceptibility at this locus" in lines 245-249 does not sound solid to the reviewer. Does the authors have promising evidence to support rs33980500 affects psoriasis risk through altering protein rather than a regulatory effect?

We appreciate the reviewer’s concern here. However, the missense variant has been previously investigated and demonstrated to confer an impairment in binding of Act1 (encoded by *TRAF3IP2*) to

other signalling molecules such as STAT3 (PMID: 30013031) and Hsp90 (PMID: 23202271) with consequent loss of Act1 function. To clarify this, we have expanded this sentence in the results section:

“We found strong evidence that the well-known psoriasis-associated missense variant rs33980500 (p.Asp10Asn; Figure 2B) – which has been shown to impair binding to signalling molecules such as STAT3³² and Hsp90³³ – may itself have a regulatory role, and independently that rs6908585 is a skin-specific regulatory variant (regulatory probability of 0.491; Figure 2C). This suggests that both coding and gene regulation of *TRAF3IP2* contribute to psoriasis susceptibility at this locus.”

8. Of the genes identified in TWAS, how many achieved significant colocalization?

Thank you for raising this important point. After performing colocalization tests with significant eQTLs (p-value threshold 1.0×10^{-5}) using GTEX v7 data and observing a lack of colocalization at several loci, we became aware of the recommendations of Barbeira et al (PMID: 29739930) who developed the S-PrediXcan TWAS approach. They suggest additional post-analysis QC steps to ensure TWAS findings are robust, specifically to: (1) select an initial list of TWAS genes based on Bonferroni correction for all gene-tissue pairs (as we had done in the submitted manuscript); (2) filter out associations that do not also have a significant prediction performance p-value after adjusting for the number of PrediXcan significant gene-tissue pairs; (3) filter out “LD-contaminated associations”, which are gene-tissue pairs for which colocalization analysis shows strong evidence for independent signals in the eQTL and primary trait data.

In light of this, we have now performed the additional filtering steps (2) and (3) with our TWAS results (full details in Methods and Supplementary Methods). This results in a reduced list of TWAS associations in which we can have much greater confidence of a role in psoriasis pathobiology: there are now 30, 43 and 29 TWAS genes identified in blood, sun-exposed skin and non-exposed skin respectively. These are presented in a revised Supplementary Table 12 (which due to renumbering replaces Supplementary Table 10 in the original submission), where we also include for additional information the strength of evidence for explicitly colocalizing signals (i.e. coloc PP4).

We note that the reduced list of TWAS genes has resulted in revised versions of Supplementary Figures 9 and 10 and Supplementary Tables 12 and 13 (describing the evidence for, and location of, the TWAS associations; due to renumbering, previously Supplementary Figures 5 and 6 and Supplementary Tables 10 and 11); Figure 3 (describing the sc-RNAseq lookup and clustering); Supplementary Figure 12 and Supplementary Table 16 (describing enrichment testing in cytokine-stimulated genes in keratinocytes; previously Supplementary Figure 9 and Supplementary Table 13); Supplementary Table 20 (summarising findings across loci; previously Supplementary Table 16); and the replacement of Supplementary Figure 8 with Supplementary Table 15 (describing functional enrichment testing for genes in each TWAS cluster). These changes do not change the broader conclusions of the manuscript.

9. For the 12 traits having potential causal relationship with psoriasis (line 327-329), have the authors evaluated the causal relationship in Mendelian randomization? If so, how many of them achieve consistent results in Mendelian randomization?

Thank you for the suggestion. We have implemented two-sample MR for all twelve traits using the same underlying GWAS summary statistics. The results are presented in Supplementary Table 19 and are broadly consistent with the findings of the LCV method. We observe that five of the twelve causal relationships to also be significant using MR in the same direction as implied by LCV, and none in the opposite direction to that implied by LCV. We note that summary statistics for the majority of the twelve

traits derive from single-study GWAS analyses in UK Biobank, and as such have relatively limited power and result in small estimated effect sizes when psoriasis is treated as the MR exposure. Overall, we conclude that the LCV approach provides estimates that are broadly consistent with MR findings, and by incorporating information outside of genome-wide significant loci, may provide increased sensitivity to detect causal relationships. In addition to relevant detail in the Methods and Supplementary Methods, we have added the following statement to the Results section:

“Five of the 12 causal relationships were observable using two-sample MR (Supplementary Table 19), the other seven highlighting the additional value of considering genome-wide variation through the GCP-based approach.”

REVIEWERS' COMMENTS

Reviewer #1 (Remarks to the Author):

Summary:

I think the authors fairly addressed the points I raised, and I am satisfied with the authors' responses. I have two minor points.

Minor comments:

1. Regarding "Effect size estimates were consistent with the previous psoriasis meta-analysis⁷ (Supplementary Figure 2)", the replicability is beautiful. I want to confirm whether there are any shared samples between the current study (Y-axis) and the 2017 meta-analysis (X-axis). If there are too many overlapped samples, this is not informative at all.

We are grateful for this feedback and have added a second panel to Supplementary Figure 2 that illustrates that effect sizes remain consistent when excluding from the meta-analysis studies that were included in the 2017 meta-analysis.

2. I have a comment regarding their response: "We thank the reviewer for this comment. We would like to clarify that the aim of our analysis was different to that of sc-linker, where the aim is to identify causal genes. Instead, we have used scRNA-seq data to observe the cellular expression distributions for the genes identified by our TWAS analysis. This process is analogous to cell marker identification, allowing us to describe expression differences across skin cell types for these TWAS genes (for which the reference panels were based on bulk RNA-seq data) and consequently to identify cell types with relatively abundant expression of TWAS target genes in psoriatic skin. We have now clarified this objective of the analysis in our Results section". I think the authors misunderstood the value of sc-linker. Sc-linker is an analytical pipeline integrating single-cell RNA-sequencing, epigenomic SNP-to-gene maps, and GWAS summary statistics to infer the underlying cell types by which genetic variants influence disease. So, sc-linker aims to identify causal cell types, not causal genes. The authors sought to identify cell types that express large amounts of TWAS-nominated genes, so their aim is similar to sc-linker. However, the difference is that sc-linker is a typical polygenic analysis (cumulative effect of genome-wide weak associations) using S-LDSC, whereas the authors' approach relied on a few loci with strong associations. Therefore, I believe the sc-linker approach is complementary to their approach and potentially enriches their depth of investigation. This is why I recommend sc-linker analysis.

We thank the reviewer for taking the time to expand on the value of sc-linker, and we agree with the point made. We have shown in our study that a relatively simple interrogation of GWAS-derived candidate genes using limited single-cell transcriptomic profiles can provide valuable mechanistic insights. But undoubtedly, this value will be enhanced in future with more comprehensive functional genomics datasets and increasingly powerful methods such as sc-linker. We appreciate that this point was not well made in our previous submission, and as such have added to the Discussion:

These efforts will benefit both from larger and higher resolution context-specific functional genomic datasets, encompassing a wider range of skin and blood cell types, and from emerging algorithms that can integrate these data with aligned genetic associations to implicate specific cell types and their contexts in the psoriasis disease process.^{78, 79}

Reviewer #3 (Remarks to the Author):

The authors have addressed all of the reviewer's concerns in the revised manuscript.

We appreciate the additional feedback from both reviewers.

Chr	LD block	With protein-altering variants
1	19705121-154	RORC
5	15110880-964	ERAP1, ERAP2
11	17298216-120	POU2F3
12	10481523-117	KLRC4
12	52647027-617	COQ10A, STAT2
16	28020777-510	AC1350481, HSD3B7, ZNF646, PRSS53
17	35285540-481	DHX58, ARHGAP27, SPPL2C, MAPT, STH, KANSL1
17	77943023-793	CARD14
19	32573487-341	PEPD
19	48509877-494	FUT2

TWAS candidates^e

TUFT1, LINGO4, THEM5, FLG, LCE3D, LCE3C, LCE3A,
LCE2A, LCE4A, C1orf68, KPRP, LCE1E, LCE1D, SPRR1B,
SPRR2D, SPRR2B

ERAP1

POU2F3

KLRC3, KLRC1

SUOX, IKZF4, RPS26, CNPY2, IL23A, STAT2, SPRYD4
RNF40, HSD3B7, STX1B, STX4, RP11-196G112,
ZNF668, ZNF646, PRSS53, VKORC1, BCKDK, KAT8,
RP11-388M209, ITGAM, ITGAX

DHX58, KCNH4, STAT3, PTRF, TUBG2, CNTNAP1,
BECN1, IFI35

CARD14, SGSH

CEBPG, PEPD

FAM83E, NTN5, FUT2, MAMSTR, RASIP1